# Iterative Scene Graph Generation

**Siddhesh Khandelwal**[1,2] **and Leonid Sigal**[1,2,3]
[1]Department of Computer Science, University of British Columbia
[2]Vector Institute for AI
[3]CIFAR AI Chair
{skhandel, lsigal}@cs.ubc.ca

## Abstract

The task of scene graph generation entails identifying object entities and their corresponding interaction predicates in a given image (or video). Due to the combinatorially large solution space, existing approaches to scene graph generation assume certain factorization of the joint distribution to make the estimation feasible (*e.g.*, assuming that objects are conditionally independent of predicate predictions). However, this fixed factorization is not ideal under all scenarios (*e.g.*, for images where an object entailed in interaction is small and not discernible on its own). In this work, we propose a novel framework for scene graph generation that addresses this limitation, as well as introduces dynamic conditioning on the image, using message passing in a Markov Random Field. This is implemented as an iterative refinement procedure wherein each modification is conditioned on the graph generated in the previous iteration. This conditioning across refinement steps allows joint reasoning over entities and relations. This framework is realized via a novel and end-to-end trainable transformer-based architecture. In addition, the proposed framework can improve existing approach performance. Through extensive experiments on Visual Genome [30] and Action Genome [25] benchmark datasets we show improved performance on the scene graph generation task. The code is available at github.com/ubc-vision/IterativeSG.

## 1 Introduction

Scene graphs allow for structured understanding of objects and their interactions within a scene. A *scene graph* is a graph where nodes represent objects within the scene, each detailed by the class label and spatial location, and the edges capture the relationships between object pairs. These relationships are usually represented by a <subject, predicate, object> triple. Effectively generating such graphs, from either images or videos, has emerged as a core problem in computer vision [14, 35, 39, 46, 50, 52, 54]. Scene graph representations can be leveraged to improve performance on a variety of complex high-level tasks like VQA [24, 47], Image Captioning [17, 53], and Image Generation [26].

The task of scene graph generation involves estimating the conditional distribution of the relationship triplets given an image. Naively modelling this distribution is often infeasible as the space of possible relationship triplets is considerably larger than the space of possible subjects, objects, and predicates. To circumvent this issue, existing methods factorize the aforementioned distribution into easy-to-estimate conditionals. For example, *two-stage* approaches, like [11, 47, 55], follow the graphical model image → [subject, object] → predicate, wherein the subjects and objects are independently obtained via a pre-trained detector like Faster-RCNN [41]. These are then consumed by a downstream network to estimate predicates. Any such factorization induces conditional dependencies (and independencies) that heavily influence model characteristics. For example, in the aforesaid graphical model, errors made during the estimation of the subjects and objects are naturally propagated towards the predicate distribution, which makes the estimation of predicates involving

classes with poor detectors challenging. Furthermore, the assumed fixed factorization might not always be optimal. Having information about predicates in an image can help narrow down the space of possible subjects/objects, *e.g.*, predicate `wearing` makes it likely that the subject is a `person`.

Additionally, due to the *two-stage* nature of most existing scene graph approaches (barring very recent methods like [14, 31, 35]), the image feature representations obtained from a pre-trained task-oblivious detector might not be optimally catered towards the scene graph generation problem. Intuitively, one can imagine that the information required to accurately localize an object *might not* necessarily be sufficient for predicate prediction, and by extension, accurate scene graph generation. Morever, two-stage approaches often suffer with efficiency issues as the detected objects are required to be paired up before predicate assignment. Doing so naively, by pairing all possible objects [50], results in quadratic number of pairs that need to be considered. Traditional approaches deal with this using heuristics, such as IoU-based threshold-based pairing [47, 55].

In this work, we aim to alleviate the previously mentioned issues arising from a fixed factorization and potentially limited object-centric feature representations by proposing a *general* framework wherein the subject, objects, and predicates can be inferred jointly (*i.e.*, depend on one another), while simultaneously avoiding the complexity of exponential search space of relational triplets. This is achieved by performing message passing within a Markov Random Field (MRF) defined by components of a relational triplet (Figure 1(a)). Unrolling this message passing is equivalent to an *iterative* refinement procedure (Figure 1(b)), where each message passing stage takes in the estimate from the previous step. Our proposed framework models this iterative refinement strategy by first producing a scene graph estimate using a traditional factorization, and then systematically improving it over multiple refinement steps, wherein each refinement is conditioned on the graph generated in the previous iteration.

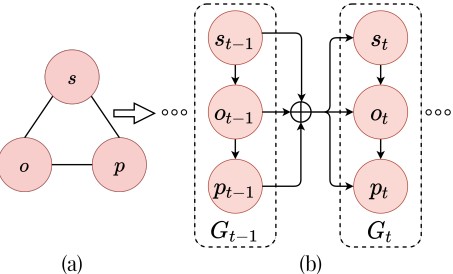

Figure 1: **Iterative Refinement for Scene Graph Generation.** By unrolling the message passing in a Markov Random Field (a), our proposed approach essentially models an iterative refinement process (b) wherein each modification step is conditioned on the previously generated graph estimate.

This conditioning across refinement steps compensates for the conditional independence assumptions and lets our framework jointly reason over the subjects ($\mathbf{s}$), objects ($\mathbf{o}$), and predicates ($\mathbf{p}$).

**Contributions.** To realize the aforementioned iterative framework, we propose a novel and intuitive transformer [48] based architecture. On a technical level, our model defines three separate multi-layer multi-output synchronized decoders, wherein each decoder layer is tasked with modeling either the subject, object, or predicate components of a relationship triplets. Therefore, the combined outputs from each layer of the three decoders generates a scene graph estimate. The inputs to each decoder layer are conditioned to enable joint reasoning across decoders and effective refinement of previous layer estimates. This conditioning is achieved *implicitly* via a novel joint loss, and *explicitly* via cross-decoder and layer-wise attention. Additionally, each decoder layer is also conditioned on the image features, which are provided by a shared encoder. As our proposed model is end-to-end trainable, it addresses the limitation of two-stage approaches, allowing image features to directly adopt to the scene graph generation task. Finally, to tackle the long-tail nature of the scene graph predicate classes [11], we employ a loss weighting strategy to enable flexible trade-off between dominant (head) and underrepresented (tail) predicate classes in the long-tail distribution. In contrast to data sampling strategies [11, 32], this has a benefit of not requiring additional fine-tuning of models with sampled data post training. We illustrate that our proposed architecture achieves state-of-the-art performance on two benchmark datasets – Visual Genome [30] and Action Genome [25]; and thoroughly analyze effectiveness of the approach as a function of the refinement steps, design choices employed and as a generic add-on to an existing, MOTIF [55], architecture.

## 2 Related Work

**Scene Graph Generation.** Scene graph generation has emerged as a popular research area in the vision community [11, 28, 31, 35, 39, 40, 46, 47, 50, 52, 54, 55]. Existing scene graph generation methods can be broadly categorized as either *one-stage* or *two-stage* approaches. The first step of the predominant approach – the *two-stage* methods – involves pre-training a strong object detector

for all object classes in the dataset, usually using detector like Faster-RCNN [41]. The graph generation network is then built on top of the object information (bounding boxes and corresponding features) obtained from the detector. This second step entails *freezing* the detector parameters and *solely* training the graph generation network. The graph generation network is realized via different architectures such as Bi-directional LSTMs [55], Tree-LSTMs [47], Graph Neural Networks [52], and other message passing algorithms [40, 50]. The limitation of all two-stage approaches is apparent – the graph generation network has no influence over the detector and object features. One stage approaches overcome this obstacle by employing architectures that allow end-to-end training, such as fully convolutional networks [35] and transformer based models [7, 14, 31, 43]. The joint optimization enables interaction between the detection and the graph generation modules, leading to better scene graphs. However, all the aforementioned methods operate under the assumption of a fixed factorized model, inducing conditional independencies that might not be ideal under all scenarios.

**Long-Tail Recognition.** The task of scene graph generation suffers from the challenge of long-tail recognition due to the combinatorial nature of relational triplets (both object and predicate distributions are skewed). Long-tail recognition, in general, is a well studied problem in literature. *Data sampling* is a popular strategy [3, 20, 21, 27, 37, 42, 58] wherein the training data is modified to either over-represent tail classes (oversampling) or under-represent the head classes (undersampling). Specific to the task of scene graph generation, [11, 32] have explored the use of data sampling strategies to improve performance on tail predicate classes. On the contrary, *Loss re-weighting* strategies [1, 9, 13, 27, 34] assign higher weights or impose larger decision boundaries for tail classes. [51] recently adopted this paradigm for the task of scene graph generation by proposing a re-weighting strategy based on correlations between predicate classes. Conceptually similar to [51], we propose a novel re-weighting strategy and illustrate its effectiveness both on its own, and in combination with data resampling.

**Transformer Models.** In [48], authors introduced a new attention-based architecture called transformers for the task of machine translation, doing away with recurrent or convolutional architectures. Transformers have since widely been adopted for a variety of tasks, such as object detection [2, 38, 57], image captioning [8, 23], and image generation [12]. More recently, scene graph generation methods have also adopted transformer architectures [7, 14, 31, 43], owing to their end-to-end trainable nature and parallelism. Perhaps conceptually closest, [14], which focuses on visual relation detection (VRD) and not scene graph prediction, leverages the composite nature of relationships to simultaneously decode an entire relationship triplet alongside its constituent subject, object, and predicate components. Unlike [14], we do not explicitly model subject-predicate-object "sum" composites, resulting in a simpler architecture, and leverage iterative refinement procedure that relies on a different factorized attention mechanism. In the latest, *concurrent* work, [31] generates a set of entities and predicates separately, and then utilizes a graph assembly procedure to match the predicates to a pair of entities. In contrast to our formulation, this precludes conditioning of entities on predicate estimates; therefore relying on accuracy of less contextualized predictions for subsequent pairing.

## 3 Formulation

For a given image $\mathbf{I}$, a scene graph $\mathbf{G}$ can be represented as a set of triplets $\mathbf{G} = \{\mathbf{r}_i\}_{i \leqslant n} = \{(\mathbf{s}_i, \mathbf{p}_i, \mathbf{o}_i)\}_{i \leqslant n}$, where $\mathbf{r}_i$ denotes the $i$-th triplet $(\mathbf{s}_i, \mathbf{p}_i, \mathbf{o}_i)$, and $n$ denotes the total number of triplets. The subject $\mathbf{s}_i$ denotes a tuple $(\mathbf{s}_{i,c}, \mathbf{s}_{i,b})$, where $\mathbf{s}_{i,c} \in \mathbb{R}^\eta$ is the one-hot class label, and $\mathbf{s}_{i,b} \in \mathbb{R}^4$ is the corresponding bounding box coordinates. $\eta$ is the total number of possible entity classes in the dataset. The object $\mathbf{o}_i$ and predicate $\mathbf{p}_i$ can similarly be represented as tuples $(\mathbf{o}_{i,c}, \mathbf{o}_{i,b})$ and $(\mathbf{p}_{i,c}, \mathbf{p}_{i,b})$ respectively. Note that, $\mathbf{p}_{i,b}$ corresponds to the box formed by the centers' of $\mathbf{s}_{i,b}$ and $\mathbf{o}_{i,b}$ as the diagonally opposite coordinates. Additionally, $\mathbf{p}_{i,c} \in \mathbb{R}^\upsilon$ represents the corresponding one-hot predicate label between the pair $(\mathbf{s}_i, \mathbf{o}_i)$, where $\upsilon$ is the total number of possible predicate classes in the dataset. Then the task of scene graph generation can be thought of as modelling the conditional distribution $\Pr(\mathbf{G} \mid \mathbf{I})$. This distribution can be expressed as a product of conditionals,

$$\Pr\left(\mathbf{G} \mid \mathbf{I}\right) = \Pr\left(\{\mathbf{s}_i\} \mid \mathbf{I}\right) \cdot \Pr\left(\{\mathbf{o}_i\} \mid \{\mathbf{s}_i\}, \mathbf{I}\right) \cdot \Pr\left(\{\mathbf{p}_i\} \mid \{\mathbf{s}_i\}, \{\mathbf{o}_i\}, \mathbf{I}\right) \qquad (1)$$

where $\{.\}$ denotes a set. For brevity, we omit explicitly mentioning the total number of set elements $n$ throughout the paper. Existing approaches model this product of conditionals by making some underlying assumptions. For example, [11, 47, 55] assume conditional independence between $\mathbf{s}_i$ and $\mathbf{o}_i$ as they rely on heuristics to obtain the entity pairs. However, modelling the conditional in Equation

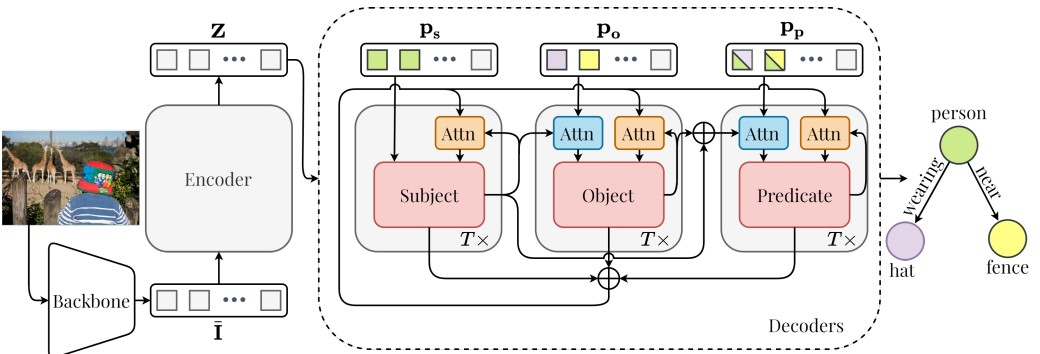

Figure 2: **Transformer Architecture for Iterative Refinement.** For a given image, the model extracts features via a convolutional backbone and a transformer encoder. The individual components of a relationship triplet are generated using separate subject, object, and predicate multi-layer decoders. The inputs to each layer of the decoder is appropriately conditioned. For example, for the predicate decoder, the positional embeddings are conditioned on the outputs generated by the subject and object decoders (blue `Attn` module) and the queries are conditioned on the previously generated graph estimate (orange `Attn` module). The model is additionally implicitly conditioned and trained in an end-to-end fashion using a joint matching loss.

1 (or any other equivalent factorization) in such a "one-shot" manner makes certain assumptions on the flow of information, which, in this case, is from $\mathbf{s}_i \to \mathbf{o}_i \to \mathbf{p}_i$. Therefore, any errors made during the estimation of $\mathbf{s}_i$ are naturally propagated towards the estimation of $\mathbf{o}_i$ and $\mathbf{p}_i$. Additionally, the subject (or object) estimation procedure $\mathrm{Pr}\left(\{\mathbf{s}_i\} \mid \mathbf{I}\right)$ is completely oblivious to the estimated predicate $\mathbf{p}_i$. Having access to such information can help the subject (or object) predictor update its beliefs and significantly narrow down the space of feasible entity pairs.

Contrary to existing works, our proposed formulation moves away from the "one-shot" generation ideology described in Equation 1. We instead argue for modelling the task of scene graph generation as an iterative refinement procedure, wherein the scene graph estimate at step $t$, $\mathbf{G}^t$, is dependent on the previous estimates $\{\mathbf{G}^{t'}\}_{t'<t}$. Formally, our aim is to model the conditional distribution $\mathrm{Pr}\left(\mathbf{G}^t \mid \{\mathbf{G}^{t'}\}_{t'<t}, \mathbf{I}\right)$. Assuming Markov property holds, this can be conveniently factorized as,

$$
\mathrm{Pr}\left(\mathbf{G}^t \mid \mathbf{G}^{t-1}, \mathbf{I}\right) = \overbrace{\mathrm{Pr}\left(\{\mathbf{s}_i^t\} \mid \mathbf{G}^{t-1}, \mathbf{I}\right)}^{\text{Subject Predictor}} \cdot \overbrace{\mathrm{Pr}\left(\{\mathbf{o}_i^t\} \mid \{\mathbf{s}_i^t\}, \mathbf{G}^{t-1}, \mathbf{I}\right)}^{\text{Object Predictor}}
$$
$$
\cdot \underbrace{\mathrm{Pr}\left(\{\mathbf{p}_i^t\} \mid \{\mathbf{s}_i^t\}, \{\mathbf{o}_i^t\}, \mathbf{G}^{t-1}, \mathbf{I}\right)}_{\text{Predicate Predictor}} .
$$

(2)

Note that even though we assume the flow of information to be from $\mathbf{s}_i^t \to \mathbf{o}_i^t \to \mathbf{p}_i^t$ for $\mathbf{G}^t$, conditioning on the previous graph estimate $\mathbf{G}^{t-1}$ allows the subject, object, and predicate predictors to jointly reason and update their beliefs, leading to better predictions. Additionally, the framework described in Equation 2 is model agnostic and can be implemented using any of the existing architectures.

## 4   Transformer Based Iterative Generation

As described in Section 3, our proposed iterative scene graph generation formulation is agnostic to the architectural choices used to realise the subject, object, and predicate predictors. In this section we provide a realization of the proposed formulation in Equation 2 using Transformer networks [48]. The choice of using transformer networks is motivated by their natural tendency to model iterative refinement behaviour under the Markov property, wherein each layer of the transformer decoder takes as input the output of the previous layer. Our novel end-to-end trainable transformer based iterative generation architecture builds on top of the DETR [2] framework, which is shown to be effective for the task of object detection. Our proposed model architecture is shown in Figure 2.

Given an image, our proposed approach first obtains corresponding image features using a combination of a convolutional backbone and a multi-layer transformer encoder, akin to DETR [2]. These image features are used as inputs to the subject, object, and predicate predictors, each implemented as

a multi-layer Transformer decoder. To generate a scene graph estimate $\widehat{\mathbf{G}}^t$ at step $t$ in accordance with Equation 2, the queries used in each predictor decoder are appropriately conditioned. For example, in the case of the predicate decoder, its input queries are conditioned on the subject ($\{\widehat{\mathbf{s}}_i^t\}$) and object ($\{\widehat{\mathbf{o}}_i^t\}$) estimates at step $t$. Additionally, the input to each predictor decoder is infused with *all* decoder estimates $\{(\widehat{\mathbf{s}}_i^{t-1}, \widehat{\mathbf{p}}_i^{t-1}, \widehat{\mathbf{o}}_i^{t-1})\}$ from the previous step $t-1$ via a structured attentional mechanism. The entire model is trained end-to-end, with a novel joint loss applied at each step $t$ to ensure the generation of a valid scene graph at every level. This section describes these components in detail.

## 4.1 Image Encoder

Similar to DETR [2], for each image $\mathbf{I}$, our proposed architecture uses a deep convolutional network (like ResNet [22]) to obtain image level spatial map $\bar{\mathbf{I}} \in \mathbb{R}^{c \times w \times h}$, where $c$ is the number of channels, and $w, h$ correspond to the spatial dimensions. A multi-layer encoder $f_e$ then transforms $\bar{\mathbf{I}}$ into a position-aware flattened image feature representation $\mathbf{Z} \in \mathbb{R}^{d \times wh}$, where $d < c$.

## 4.2 Predictor Decoders

Our approach models each of the subject, object, and predicate predictors using a multi-layer transformer decoder [2], which are denoted by $f_{\mathbf{s}}$, $f_{\mathbf{o}}$, and $f_{\mathbf{p}}$ respectively. The $t$-th layer of each decoder, denoted as $f_{\mathbf{x}}^t; \mathbf{x} \in \{\mathbf{s}, \mathbf{o}, \mathbf{p}\}$, is tasked with generating the step $t$ scene graph $\mathbf{G}^t$. Therefore, at each step $t$, the decoders output a set of $n$ feature representations $\{\mathbf{q}_{\mathbf{x},i}^t\}; \mathbf{x} \in \{\mathbf{s}, \mathbf{o}, \mathbf{p}\}$, which are transformed into a set of triplet estimates $\{(\widehat{\mathbf{s}}_i^t, \widehat{\mathbf{p}}_i^t, \widehat{\mathbf{o}}_i^t)\}$ via fully-connected feed forward layers.

Specifically, for a decoder $f_{\mathbf{x}}; \mathbf{x} \in \{\mathbf{s}, \mathbf{o}, \mathbf{p}\}$, an arbitrary layer $t$ takes as input a set of queries $\{\mathbf{q}_{\mathbf{x},i}^{t-1}\}$ and a set of learnable positional encodings $\{\mathbf{p}_{\mathbf{x},i}\}$, where $\mathbf{q}_{\mathbf{x},i}^{t-1}; \mathbf{p}_{\mathbf{x},i} \in \mathbb{R}^d$. The output representations $\{\mathbf{q}_{\mathbf{x},i}^t\}$ are then obtained via a combination of self-attention between the input queries, and encoder-decoder attention across the encoder output $\mathbf{Z}$. These attention modules allow the decoder to jointly reason across all queries, while simultaneously incorporating context from the input image.

At a given step $t$, naively using the inputs $\{\mathbf{q}_{\mathbf{x},i}^{t-1}\}$ and $\{\mathbf{p}_{\mathbf{x},i}\}$ to generate $\mathbf{G}^t$ forgoes leveraging the compositional property of relations. As described Equation 2, for any arbitrary step $t$, our proposed formulation entails *two* types of conditioning for better scene graph estimation. The first involves conditioning decoders on the step $t$ outputs, specifically the object decoder $f_{\mathbf{o}}^t$ on the subject decoder $f_{\mathbf{s}}^t$, and the predicate decoder $f_{\mathbf{p}}^t$ on both $f_{\mathbf{s}}^t$ and $f_{\mathbf{o}}^t$. The second requires all three decoder layers at step $t$ to be conditioned on the outputs generated at step $t-1$. To effectively implement this design, we modify the inputs to each decoder layer $f_{\mathbf{x}}^t$. Specifically, the positional encodings $\{\mathbf{p}_{\mathbf{x},i}\}$ are modified to condition them on step $t$ outputs, and the queries $\mathbf{q}_{\mathbf{x},i}^{t-1}$ are updated to incorporate information from the previous step $t-1$. Modifying the positional encoding and queries separately allows the model to easily disentangle and differentiate between the two conditioning types.

**Conditional Positional Encodings.** At a particular step $t$, the conditional positional encodings for the three decoders are obtained as,

$$\{\widehat{\mathbf{p}}_{\mathbf{s},i}^t\} = \{\mathbf{p}_{\mathbf{s},i}\}; \quad \{\widehat{\mathbf{p}}_{\mathbf{o},i}^t\} = \{\mathbf{p}_{\mathbf{o},i}\} + \texttt{FFN}\left(\texttt{MultiHead}\left(\{\mathbf{p}_{\mathbf{o},i}\}, \{\widetilde{\mathbf{q}}_{\mathbf{s},i}^t\}, \{\mathbf{q}_{\mathbf{s},i}^t\}\right)\right)$$
$$\{\widehat{\mathbf{p}}_{\mathbf{p},i}^t\} = \{\mathbf{p}_{\mathbf{p},i}\} + \texttt{FFN}\left(\texttt{MultiHead}\left(\{\mathbf{p}_{\mathbf{p},i}\}, \{\widetilde{\mathbf{q}}_{\mathbf{s},i}^t \oplus \widetilde{\mathbf{q}}_{\mathbf{o},i}^t\}, \{\mathbf{q}_{\mathbf{s},i}^t \oplus \mathbf{q}_{\mathbf{o},i}^t\}\right)\right)$$

(3)

where $\texttt{MultiHead(Q, K, V)}$ is the Multi-Head Attention module introduced in [48], $\texttt{FFN(.)}$ is a fully-connected feed forward network, and $\oplus$ is the concatenation operation. Additionally, $\widetilde{\mathbf{q}}_{\mathbf{x},i}^t = \mathbf{q}_{\mathbf{x},i}^t + \mathbf{p}_{\mathbf{x},i}; \mathbf{x} \in \{\mathbf{s}, \mathbf{o}, \mathbf{p}\}$ is the position-aware query.

**Conditional Queries.** Similarly, for a step $t$, conditional queries for the subject decoder are defined,

$$\{\widehat{\mathbf{q}}_{\mathbf{s},i}^{t-1}\} = \{\mathbf{q}_{\mathbf{s},i}^{t-1}\} + \texttt{FFN}\left(\texttt{MultiHead}\left(\{\widetilde{\mathbf{q}}_{\mathbf{s},i}^{t-1}\}, \{\widetilde{\mathbf{q}}_{\mathbf{s},i}^{t-1} \oplus \widetilde{\mathbf{q}}_{\mathbf{o},i}^{t-1} \oplus \widetilde{\mathbf{q}}_{\mathbf{p},i}^{t-1}\}, \{\mathbf{q}_{\mathbf{s},i}^{t-1} \oplus \mathbf{q}_{\mathbf{o},i}^{t-1} \oplus \mathbf{q}_{\mathbf{p},i}^{t-1}\}\right)\right)$$

(4)

$\widehat{\mathbf{q}}_{\mathbf{o},i}^{t-1}$ and $\widehat{\mathbf{q}}_{\mathbf{p},i}^{t-1}$ are defined identically. For a decoder layer $f_{\mathbf{x}}^t; \mathbf{x} \in \{\mathbf{s}, \mathbf{o}, \mathbf{p}\}$ we use the conditioned positional encodings $\{\widehat{\mathbf{p}}_{\mathbf{x},i}^t\}$ and queries $\{\widehat{\mathbf{q}}_{\mathbf{x},i}^{t-1}\}$ as input.

## 4.3 End-To-End Learning

Our proposed transformer based refinement architecture can be trained in an end-to-end fashion. To ensure that a valid scene graph is generated at every level, we propose a novel joint loss that is applied at each step $t$. Therefore, the combined loss $\mathcal{L}$ can be expressed as $\mathcal{L} = \sum_t \mathcal{L}^t = \sum_t \mathcal{L}_{\mathbf{s}}^t + \mathcal{L}_{\mathbf{o}}^t + \mathcal{L}_{\mathbf{p}}^t$,

where $\mathcal{L}_{\mathbf{x}}^t; \mathbf{x} \in \{\mathbf{s}, \mathbf{o}, \mathbf{p}\}$ represents the loss applied to the $t$-th layer of the decoder $f_{\mathbf{x}}$. Our approach generates a fixed-size set of $n$ triplet estimates $\{\widehat{\mathbf{r}}_i^t\} = \{(\widehat{\mathbf{s}}_i^t, \widehat{\mathbf{p}}_i^t, \widehat{\mathbf{o}}_i^t)\}$ at each step $t$, where $n$ is larger than the number of ground truth relations for a given image. Therefore, in order to effectively optimize the proposed model, we obtain an optimal bipartite matching between the predicted and ground truth triplets. Note that, contrary to the matching algorithm in [2], our proposed matching is defined over triplets rather than individual entities. Additionally, instead of independently computing the loss over each decoder layer as in [2], our loss computes a joint matching across all refinement layers.

Let $\mathbf{G} = \{\mathbf{r}_i\} = \{(\mathbf{s}_i, \mathbf{p}_i, \mathbf{o}_i)\}$ denote the ground truth scene graph for an image $\mathbf{I}$. Note that, as the number of ground truth relations is less than $n$, we convert $\mathbf{G}$ to a $n$-sized set by padding it with $\varnothing$ (no relation). The goal then is to find a bipartite matching between the ground truth graph $\mathbf{G}$ and the set of all graph estimates $\{\widehat{\mathbf{G}}^t\}$ that minimizes the *joint matching cost*. Specifically, assuming $\sigma$ to be a valid permutation of $n$ elements,

$$\widehat{\sigma} = \arg\min_{\sigma} \sum_t \sum_i^n \mathcal{L}_{\text{rel}}\left(\mathbf{r}_i, \widehat{\mathbf{r}}_{\sigma(i)}^t\right) \tag{5}$$

where the pair-wise relation matching cost $\mathcal{L}_{\text{rel}}$ is defined as,

$$\mathcal{L}_{\text{rel}}\left(\mathbf{r}_i, \widehat{\mathbf{r}}_{\sigma(i)}^t\right) = -\mathbb{1}_{\{\mathbf{r}_i \neq \varnothing\}}\begin{bmatrix} \widehat{\mathbf{s}}_{\sigma(i),c}^t \cdot \mathbf{s}_{i,c} - \mathcal{L}_{\text{box}}\left(\widehat{\mathbf{s}}_{\sigma(i),b}^t, \mathbf{s}_{i,b}\right) \\ + \widehat{\mathbf{o}}_{\sigma(i),c}^t \cdot \mathbf{o}_{i,c} - \mathcal{L}_{\text{box}}\left(\widehat{\mathbf{o}}_{\sigma(i),b}^t, \mathbf{o}_{i,b}\right) \\ + \widehat{\mathbf{p}}_{\sigma(i),c}^t \cdot \mathbf{p}_{i,c} - \mathcal{L}_{\text{box}}\left(\widehat{\mathbf{p}}_{\sigma(i),b}^t, \mathbf{p}_{i,b}\right) \end{bmatrix} \tag{6}$$

where $\cdot$ is the vector dot product, and $\mathcal{L}_{\text{box}}$ is a combination of the L-1 and generalized IoU losses. Please refer to Section 3 for clarification on the notations. The optimal permutation $\widehat{\sigma}$ can then be computed using the Hungarian algorithm. The loss $\mathcal{L}_{\mathbf{s}}^t$ is then defined as,

$$\mathcal{L}_{\mathbf{s}}^t = \sum_{i=1}^n \left[ -\log\left(\widehat{\mathbf{s}}_{\widehat{\sigma}(i),c}^t \cdot \mathbf{s}_{i,c}\right) + \mathbb{1}_{\{\mathbf{r}_i \neq \varnothing\}} \mathcal{L}_{\text{box}}\left(\widehat{\mathbf{s}}_{\widehat{\sigma}(i),b}^t, \mathbf{s}_{i,b}\right) \right] \tag{7}$$

$\mathcal{L}_{\mathbf{o}}^t$ and $\mathcal{L}_{\mathbf{p}}^t$ are defined identically. Note that as we use the same permutation $\widehat{\sigma}$ for all refinement layers $t$, it induces strong *implicit* dependencies between the subject, object, and predicate decoders. The potency of the aforementioned implicit conditioning is highlighted in the experiment section.

### 4.4 Loss Re-Weighting

Due to the inherent long-tail nature of the scene graph generation task, using an unbiased loss often leads to the model prioritizing the most common (*a.k.a.,* head) predicate classes like `has` and `on`, which have abundant training examples. To afford our proposed model the flexibility to achieve the trade-off between head/tail classes, we integrate a loss-reweighing scheme into the model training procedure. Note that, contrary to existing methods that do this post-hoc via finetuning the final layer of the trained network (see Section 2), we instead train the model with this weighting to allow the internal feature representations to reflect the desired trade-off. Note, to the best of our knowledge, our paper is the first to illustrate effectiveness of such a strategy for the task of scene graph generation. For a particular predicate class $c \in [1, v]$, we define the class weight $w_c$ as $\max\left\{(\alpha/f_c)^\beta, 1.0\right\}$, where $f_c$ is the frequency of the predicate class $c$ in the training set, and $\{\alpha, \beta\}$ are scaling parameters. Note that this weighting scheme is similar to the data sampling strategy described in [19, 32]. However, instead of modifying the training set, we scale each class weight by the factor $w_c$ when computing the predicate classification loss $\mathcal{L}_{\mathbf{p}}^t$. Therefore, $\mathcal{L}_{\mathbf{p}}^t$ can be defined similarly to Equation 7,

$$\mathcal{L}_{\mathbf{p}}^t = \sum_{i=1}^n \left[ -w_c \log\left(\widehat{\mathbf{p}}_{\widehat{\sigma}(i),c}^t \cdot \mathbf{p}_{i,c}\right) + \mathbb{1}_{\{\mathbf{r}_i \neq \varnothing\}} \mathcal{L}_{\text{box}}\left(\widehat{\mathbf{p}}_{\widehat{\sigma}(i),b}^t, \mathbf{p}_{i,b}\right) \right]. \tag{8}$$

## 5 Experiments

We demonstrate the effectiveness of our proposed approach on two datasets,

**Visual Genome [30].** This is a benchmark for scene graph generation. We use the common processed subset from [50], which contains $108k$ images, with 150 object and 50 predicate categories.

**Action Genome [25].** This dataset provides scene graph annotations over videos in the Charades dataset [44] for the task of human-object interaction. It contains $9,848$ videos across 35 object

Table 1: **Scene Graph Generation on Visual Genome.** Mean Recall (mR@K), Recall (R@K), and Harmonic Recall (hR@K) shown for baselines and our approach. **B**=Backbone, **D**=Detector. Baseline results are borrowed from [31]. $M$ indicates the number of top-k links used by the baseline [31]. Note that our approach implicitly assumes $M = 1$. For the approaches that use the ResNet-101 backbone [22] (akin to our method), the best result is highlighted in red, second best in blue.

| B | D | # | Method | mR@50/100 | R@50/100 | hR@50/100 | Head | Body | Tail |
|---|---|---|--------|-----------|----------|-----------|------|------|------|
| X101-FPN | Faster RCNN | ① | RelDN [56] | 6.0 / 7.3 | 31.4 / 35.9 | 10.1 / 12.1 | - | - | - |
| | | ② | MOTIF [55] | 5.5 / 6.8 | 32.1 / 36.9 | 9.4 / 11.5 | - | - | - |
| | | ③ | VCTree [47] | 6.6 / 7.7 | 31.8 / 36.1 | 10.9 / 12.7 | - | - | - |
| | | ④ | BGNN [32] | 10.7 / 12.6 | 31.0 / 35.8 | 15.9 / 18.6 | 34.0 | 12.9 | 6.0 |
| | | ⑤ | VCTree-TDE [46] | 9.3 / 11.1 | 19.4 / 23.2 | 12.6 / 15.0 | - | - | - |
| | | ⑥ | VCTree-DLFE [6] | 11.8 / 13.8 | 22.7 / 26.3 | 15.5 / 18.1 | - | - | - |
| | | ⑦ | VCTree-EBM [45] | 9.7 / 11.6 | 20.5 / 24.7 | 13.2 / 15.8 | - | - | - |
| | | ⑧ | VCTree-BPLSA [18] | 13.5 / 15.7 | 21.7 / 25.5 | 16.6 / 19.4 | - | - | - |
| | | ⑨ | DT2-ACBS [11] | 22.0 / 24.4 | 15.0 / 16.3 | 17.8 / 19.5 | - | - | - |
| ResNet-101 | | ⑩ | BGNN [32, 31] | 8.6 / 10.3 | 28.2 / 33.8 | 13.2 / 15.8 | 29.1 | 12.6 | 2.2 |
| | | ⑪ | RelDN [56, 31] | 4.4 / 5.4 | 30.3 / 34.8 | 7.7 / 9.3 | 31.3 | 2.3 | 0.0 |
| | DETR | ⑫ | AS-Net [4] | 6.1 / 7.2 | 18.7 / 21.1 | 9.2 / 10.7 | 19.6 | 7.7 | 2.7 |
| | | ⑬ | HOTR [29] | 9.4 / 12.0 | 23.5 / 27.7 | 13.4 / 16.7 | 26.1 | 16.2 | 3.4 |
| | | | Concurrent Work | | | | | | |
| | | ⑭ | SGTR$_{M=1}$ [31] | 12.0 / 14.6 | 25.1 / 26.6 | 16.2 / 18.8 | 27.1 | 17.2 | 6.9 |
| | | ⑮ | SGTR$_{M=3}$ [31] | 12.0 / 15.2 | 24.6 / 28.4 | 16.1 / 19.8 | 28.2 | 18.6 | 7.1 |
| | | ⑯ | SGTR$_{M=3,\text{BGNN [32]}}$ [31] | 15.8 / 20.1 | 20.6 / 25.0 | 17.9 / 22.3 | 21.7 | 21.6 | 17.1 |
| | | ⑰ | Ours$_{(\alpha=0.0,\beta=*)}$ | 8.0 / 8.8 | 29.7 / 32.1 | 12.6 / 13.8 | 31.7 | 9.0 | 1.4 |
| | | ⑱ | Ours$_{(\alpha=0.14,\beta=0.5)}$ | 14.4 / 16.4 | 27.9 / 30.4 | 19.0 / 21.3 | 30.0 | 17.3 | 11.2 |
| | | ⑲ | Ours$_{(\alpha=0.07,\beta=0.75)}$ | 15.7 / 17.8 | 27.2 / 29.8 | 19.9 / 22.3 | 28.5 | 18.8 | 13.3 |
| | | ⑳ | Ours$_{(\alpha=0.14,\beta=0.75)}$ | 15.8 / 18.2 | 26.1 / 28.7 | 19.7 / 22.3 | 28.2 | 19.4 | 13.8 |
| | | ㉑ | Ours$_{(\alpha=0.14,\beta=0.75),\text{BGNN [32]}}$ | 17.1 / 19.2 | 22.9 / 25.7 | 19.6 / 22.0 | 24.4 | 20.2 | 16.4 |
| | | ㉒ | Ours$_{(\alpha=0.14,\beta=0.75),M=3}$ | 19.5 / 23.4 | 30.8 / 35.6 | 23.9 / 28.2 | 32.9 | 28.1 | 15.8 |

(excluding class `person`) and 25 relation categories. As not all video frames are annotated, consistent with prior work [16], we use the annotated frames provides by [25]. Additionally, as in [16], we remove frames without any `person` or object annotations as they do not provide usable scene graphs.

**Implementation Details (***transformer-based approach***).** We use ResNet-101 [22] as the backbone network for image feature extraction. Each of the subject, object, and predicate decoders have 6 layers, with a feature size of 256. The decoders use 300 queries. For training we use a batch size of 12 and initial learning rate of $10^{-4}$, which is gradually decayed. Note that although our model predicts individual relation triplets, a graph is obtained by applying a non-maximum suppression (NMS) strategy [15, 41] to group the predicted subjects and objects into entity instances.

**Implementation Details (***MOTIF***).** For the re-implementation of the MOTIF [55] baseline and the subsequent augmentation of our proposed framework, we follow the same training procedure as [55]. Specifically, we assume the Faster-RCNN detector [41] with the ResNeXt-101-FPN [49] backbone. The detector is first pre-trained on the Visual Genome dataset [30]. As is the case with two-stage approaches, when learning the scene graph generator, the detector parameters are frozen. The specifics on how our approach is augmented to MOTIF is described in the supplementary (Sec. A).

**Evaluation Metrics.** To measure performance, we report results using standard scene graph evaluation metrics, namely Recall (R@K) [50] and Mean Recall (mR@K) [5, 47]. While recall is class agnostic, mean recall averages the recalls computed for each predicate category independently. Usually, higher R@K is indicative of better performance on dominant (head) classes, where as higher mR@K suggests better tail class performance. Recent methods [11, 32] have argued for the use of mean recall as it reduces influence of dominant relationships such as on and `has` on the metric. However, even for the same model architecture, one can trade-off R@K for mR@K by using long tail recognition techniques described in Section 2. This trade-off is seldom analyzed or reported, making it difficult to compare performance across methods. Therefore, inspired by the generalized zero-shot

Table 2: **Ablation of Model Components.** Mean Recall (mR@K) and Recall (R@K) reported on the Visual Genome dataset. **CAS**=Conditioning Across Steps (Eq. 4), **CWS**=Conditioning Within a Step (Eq. 3), **JL**=Joint Loss (Sec. 4.3).

| # | CAS | CWS | JL | mR@20/50 | R@20/50 |
|---|---|---|---|---|---|
| ① | ✓ | ✓ | ✓ | **11.8 / 15.8** | **21.0 / 26.1** |
| ② | | | | 1.7 / 1.9 | 2.7 / 4.3 |
| ③ | | | ✓ | 11.2 / 14.7 | 20.1 / 25.4 |
| ④ | | ✓ | ✓ | 11.5 / 15.3 | 20.9 / 26.1 |
| ⑤ | ✓ | | ✓ | 11.7 / 15.4 | 20.9 / 25.9 |

Table 3: **Augmentation to MOTIF.** Mean Recall (mR@K) and Recall (R@K) is reported on the Visual Genome dataset for the baseline and the variant with our iterative formulation. † denotes our re-implementation of the method.

| Model | t | mR@20/50 | R@20/50 |
|---|---|---|---|
| MOTIF† [55] | | 6.0 / 8.0 | 23.6 / 30.4 |
| MOTIF† | 1 | 6.0 / 8.1 | 23.7 / 30.6 |
| + | 2 | 6.2 / 8.4 | 23.9 / 30.7 |
| Ours | 3 | **6.4 / 8.5** | **24.0 / 30.8** |

Table 4: **Ablation of Loss Re-weighting Parameters.** We vary $\alpha, \beta$ (Sec. 4.4) and report recall, mean recall, and harmonic recall on the Visual Genome test set.

| $\alpha$ | $\beta$ | mR@20/50 | R@20/50 | hR@20/50 |
|---|---|---|---|---|
| 0.0 | * | 5.8 / 8.0 | **24.2 / 29.7** | 9.4 / 12.6 |
| 0.07 | 0.75 | 11.2 / 15.7 | 21.8 / 27.2 | 14.8 / 19.9 |
| 0.14 | 0.75 | **11.8 / 15.8** | 21.0 / 26.1 | **15.1 / 19.7** |
| 0.21 | 0.75 | 11.3 / 15.5 | 20.0 / 24.8 | 14.4 / 19.1 |
| 0.14 | 0.5 | 10.3 / 14.4 | **22.6 / 27.9** | 14.2 / 19.0 |
| 0.14 | 0.625 | 11.0 / 15.1 | 21.4 / 26.4 | 14.5 / 19.2 |
| 0.14 | 0.75 | **11.8 / 15.8** | 21.0 / 26.1 | **15.1 / 19.7** |
| 0.14 | 0.875 | 11.2 / **15.9** | 19.3 / 24.5 | 14.2 / 19.3 |

Table 5: **Ablation of Model Parameters.** ① = Our proposed transformer approach with 6 decoder layers. ② = transformer with 6 decoder layers devoid of our proposed refinement. ③ = transformer with 1 decoder layer devoid of our proposed refinement.

| # | mR@20/50 | R@20/50 | Model Size |
|---|---|---|---|
| ① | **11.8 / 15.8** | **21.0 / 26.1** | 1× |
| ② | 1.7 / 1.9 | 2.7 / 4.3 | 0.9× |
| ③ | 9.9 / 13.1 | 17.5 / 21.8 | 1.1× |

learning harmonic average metric [10], we propose a new metric – *harmonic recall* (hR@K), defined as the harmonic mean of mR@K and R@K. Such a metric encourages methods to have a healthy balance between the head and tail class performance. Additionally, we report the performance of our model for different values of the loss weighing parameters described in Section 4.4. Furthermore, we also report the mR@100 on each long-tail category subset, namely *head*, *body*, and *tail*, as in [32].

## 5.1 Ablation Study

All the subsequent ablation studies are done on the Visual Genome dataset [30].

**Model Components.** We analyze the importance of each of our model components, namely the conditioning within a particular step $t$ (CWS; Equation 3), the conditioning across steps (CAS; Equation 4), and the joint loss (JL; Section 4.3) in Table 2. For a fair comparison, all models are trained using the *same* loss weighing parameters, $\alpha = 0.14, \beta = 0.75$. It can be seen that our proposed novel joint loss provides a significant improvement in performance (③) when compared to the ablated model that uses separate losses (as in [2]; ②). This is largely due to the joint loss being able to induce strong implicit dependencies between the three decoders. As a consequence, even without CAS and CWS, the proposed joint loss by itself enables refinement. The two forms of conditioning (④-⑤) provide a structured pathway to incorporating knowledge from previously generated estimates or other triplet components in an *explicit* fashion, leading to improved scene graphs. CAS (⑤) builds upon the joint loss, integrating information that the aforementioned implicit conditioning is unable to capture. CWS (④) is complementary to the refinement process, and allows for more consistent graph generation within a step. As these two types of conditionings capture complementary information, using them together (①) further improves on performance.

**Refinement Across Steps.** To further analyze the effectiveness of our refinement procedure, we contrast the quality of scene graphs generated at each refinement step for a particular trained model ($\alpha = 0.14, \beta = 0.75$). We provide a detailed analysis in the supplementary (Table A1). Visually, Figure 3 highlights steady improvements in the graph quality over refinement steps. Owing to our structured conditioning, the model is able to improve on both predicate detection and entity localization over refinement iterations.

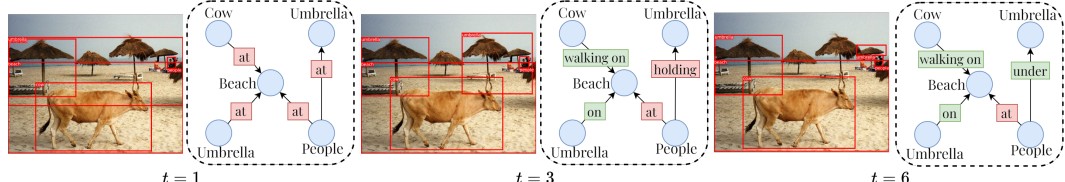

Figure 3: **Qualitative Refinement Analysis.** Graph estimates for different refinement steps shown. Colours red and green indicate incorrect and correct predictions respectively. Additional visualizations are shown in the supplementary (Section C.3).

**Loss Re-Weighting.** As mentioned in Section 4.4, we employ a simple loss re-weighting scheme, allowing for an easy trade-off between recall for mean recall. Some results with different weightings are reported in Table 1 (⑰–⑳). To investigate this further, we train our proposed model with different values for the parameters $\alpha$ and $\beta$, and report the performance in Table 4. We fix $\beta$ in the top half of the table (first 4 rows), and $\alpha$ in the bottom half (last 4 rows). It can be seen that increasing $\alpha$ and $\beta$ generally biases the model more towards the tail classes (higher mean recall), whereas lower values tend to favour the head classes (higher recall). As a consequence, our proposed approach is flexibly able to function in a wide range of recall and mean recall values. However, as is the case with every long-tail recognition approach, choosing values that are too high (*e.g.* $\alpha = 0.21, \beta = 0.75$) lead to poor performance due to the model being extremely biased and therefore unable to learn accurate feature representations on the popular classes.

**Model Parameters.** We additionally argue that the improvements obtained by our proposed refinement procedure are not a direct consequence of having an increased number of parameters. To corroborate this, we contrast our proposed model with alternatives with similar number of parameters but devoid of refinement in Table 5. ① corresponds to our transformer-based approach with 6 decoder layers that uses the proposed joint loss and both forms of conditioning. ② is a similar model with 6 decoder layers but bereft of the aforementioned proposed implicit and explicit conditionings. ③, on the other hand, is a *single* decoder layer transformer model without refinement, wherein the number of parameters are increased to be comparable to our proposed approach. The inferior performance of ② and ③ highlight that increasing model capacity either via making each layer larger or via adding more layers (increasing model depth) does not emulate refinement. Our proposed refinement procedure, realized by the joint loss and explicit conditionings, allows for better learning, and the performance gains observed cannot be attributed to having additional parameters.

## 5.2 Comparison to Existing Methods

**Visual Genome.** We report the mean recall, recall, and harmonic recall values contrasting our proposed model with existing scene graph methods, including the concurrent work in [31], in Table 1. Contrary to existing approaches, depending on the loss weighting parameters used, our proposed approach is able to easily operate on a wide spectrum of recall and mean recall values (⑰-⑳). More concretely, compared to the most competitive baseline in SGTR [31] (⑭-⑯) and other one stage methods in AS-Net [4] (⑫) and HOTR [29] (⑬), our approach is able to achieve a considerably better recall on the *head* classes (⑰; 3.5 mR@100 better on head classes compared to [31]). Additionally, when comparing a variant of our approach that has a similar R@100 value to SGTR (⑮ and ⑳), we do significantly better on mean recall (3.8 higher mR@50) highlighting the proficiency of our method to generalize to *tail* classes wherein the training data is limited. Furthermore, our proposed model in ⑳ achieves the best performance across all models (including two-stage methods that use a superior backbone [49]) on hR@50/100, underlining the capability of our method to perform well on both the head and tail classes *simultaneously*. SGTR [31] additionally reports numbers by utilizing a data sampling approach BGNN [32] to bias the model towards the tail classes (⑯). We highlight that, compared to ⑯, our model in ⑳ performs similarly on mR@50 but much better R@50/100 (5.5 R@50 higher). The poorer performance on mR@100 is largely due to SGTR [31] selecting top-3 subjects and objects for each predicate, leading to more relationship triplets being generated. On the contrary, our approach only uses 1 predicate per subject-object pair. For a fairer comparison, we evaluate our model in ⑳ using a similar top-$k$ strategy, wherein we select the top-3 predicates for each subject-object pair. The resulting model (㉒) outperforms the closest baseline by a significant margin (3.3 higher mR@100, 10.6 higher R@100 compared to ⑯). Additionally, we show that our loss re-weighting strategy is complementary to existing data sampling approaches by fine-tuning our trained model in ⑳ using BGNN [32] (㉑). For further analysis, see supplementary Section C.2.

Table 6: **Scene Graph Prediction on Action Genome.** Mean Recall (mR@K), Recall (R@K), and Harmonic Recall (hR@K) shown for baselines and variants of our approach with different loss weighting parameters. **B**=Backbone, **D**=Detector. Baseline results are taken from [16].

| B | D | # | Method | R@20 / 50 | mR@20/ 50 | hR@20/50 |
|---|---|---|--------|-----------|-----------|----------|
| X101-FPN | Faster RCNN | ① | VRD [36] | 10.3 / 10.9 | - | |
| | | ② | Freq Prior [55] | 24.0 / 24.9 | - | - |
| | | ③ | IMP [50] | 23.9 / 25.5 | - | - |
| | | ④ | MSDN [33] | 24.0 / 25.6 | - | - |
| | | ⑤ | Graph R-CNN [52] | 24.1 / 25.8 | - | - |
| | | ⑥ | RelDN [56] | 25.0 / 26.2 | - | - |
| | | ⑦ | SimpleBase [16] | 27.9 / 30.4 | 8.3 / 9.1 | 12.8 / 14.0 |
| R-101 | DETR | ⑧ | Ours$_{\{\alpha=0, \beta=*\}}$ | **31.0** / **37.5** | 20.9 / 25.3 | 24.9 / 30.2 |
| | | ⑨ | Ours$_{\{\alpha=0.07, \beta=0.75\}}$ | 30.1 / 36.5 | 36.9 / 42.8 | **33.1** / **39.4** |
| | | ⑩ | Ours$_{\{\alpha=0.14, \beta=0.75\}}$ | 29.2 / 35.3 | **37.9** / **44.0** | 32.9 / 39.2 |

**Action Genome.** We additionally report recall, mean recall, and harmonic recall values on the Action Genome dataset [25] in Table 6. Similar to the observations on Visual Genome, we observe 3.1 higher R@20 and 12.6 higher mR@20 (⑧) when compared to the closest baseline in [16] (⑦) despite using an inferior backbone [49]. Furthermore, by effectively biasing our model towards the *tail* classes, we are able to achieve 29.6 better mR@20 while having better R@20/50 values (⑩) compared to ⑦.

We provide per class relation recall and object AP for both datasets in the supplementary (Section C).

### 5.3 Generality of Proposed Formulation

Although our transformer model effectively generates better scene graphs, the proposed iterative refinement formulation (Equation 2) can be applied to any existing method. To demonstrate this, we augment this framework to the simple two-stage MOTIF [55] architecture. The details on how this is achieved is deferred to the supplementary (Section A). We contrast the performance of the baseline and our augmented approach on the Visual Genome dataset [30] in Table 3. It can be seen that with each refinement step, the model is able to generate better scene graphs (∼6% higher on mR@20 and mR@50 after 3 steps). Note that, owing to the two-stage nature of MOTIF [55], the image features used by our refinement framework are fixed and oblivious to the task of scene graph generation (as the detector parameters are frozen). Contrary to the proposed transformer model that can query the image at each layer, the performance gains when using the augmented MOTIF variant are largely limited due to the image features being the same across the refinement steps.

## 6 Conclusion, Limitations, and Societal Impact

In this work we propose a novel and general framework for iterative refinement that overcomes limitations arising from a fixed factorization assumption in existing methods. We demonstrate its effectiveness through a transformer based end-to-end trainable architecture.

**Limitations.** Limitations of the proposed approach include using a shared image encoder, which improves efficiency, but may limit diversity of representations available for different decoder layers; and limited ability to model small objects inherited from the DETR object detection design [31].

**Societal Impact.** While our model does not have any direct serious societal implications, its impact could be consequential if it is used in specific critical applications (*e.g.*, autonomous driving). In such cases both the overall performance and biases in object / predicate predictions would require careful analysis and further calibration. Our loss re-weighting strategy is a step in that direction.

## Acknowledgments and Disclosure of Funding

This work was funded, in part, by the Vector Institute for AI, Canada CIFAR AI Chair, NSERC CRC and an NSERC DG and Accelerator Grants. Hardware resources used in preparing this research were provided, in part, by the Province of Ontario, the Government of Canada through CIFAR, and companies sponsoring the Vector Institute[1]. Additional support was provided by JELF CFI grant and Compute Canada under the RAC award. Finally, we sincerely thank Bicheng Xu and Muchen Li for their valuable feedback on the paper draft.

---

[1]www.vectorinstitute.ai/#partners

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
