# Supplementary: Iterative Scene Graph Generation

**Siddhesh Khandelwal**[1,2] **and Leonid Sigal**[1,2,3]
[1]Department of Computer Science, University of British Columbia
[2]Vector Institute for AI
[3]CIFAR AI Chair
{skhandel, lsigal}@cs.ubc.ca

The code for our approach is available at github.com/ubc-vision/IterativeSG.

## A    Augmenting Iterative Framework to MOTIF

As described in Section 3 of the main paper, our proposed iterative refinement formulation is model agnostic and can be integrated to any existing architecture. We demonstrate this in Section 5.3 of the main paper by augmenting our proposed approach to an existing model in MOTIF [55], showing improved scene graph generation performance. In this section we present the details of this integration.

As is the case with most two-stage architectures, MOTIF [55] relies on a pre-trained detector (like Faster R-CNN [41]) to generate bounding box proposals $\mathbf{B} = \{\mathbf{b}_i\}$ and corresponding labels $\mathbf{L} = \{\mathbf{l}_i\}$ for entities $\mathbf{E} = \{\mathbf{e}_i\}$ within a given image. For each of these bounding boxes, the model extracts corresponding image features $\mathbf{z}_i$ via a Region of Interest (ROI) Pooling [41] operation. Additionally, as the task of scene graph prediction requires a pair of entities to associate a relation with, an IOU-based thresholding is used to couple entities into candidate pairs $\{(\mathbf{e}_i, \mathbf{e}_j)\}$. Subsequently, for each of these entity pairs $(\mathbf{e}_i, \mathbf{e}_j)$, features representing the union of their corresponding bounding boxes $(\mathbf{b}_i, \mathbf{b}_j)$ is extracted via a similar ROI pooling operation. We represent these union features as $\mathbf{u}_{i,j}$.

MOTIF [55] utilizes the set of features $\{\mathbf{z}_i\}$ and $\{\mathbf{u}_{i,j}\}$, alongside the bounding box proposals $\{\mathbf{b}_i\}$ and entity label estimates $\{\mathbf{l}_i\}$, to generate object/subject labels and the corresponding predicate between them. MOTIF achieves this by instantiating entity $f_{\mathbf{s},\mathbf{o}}$ and predicate $f_{\mathbf{p}}$ networks, each utilizing bidirectional LSTM. Note that, contrary to our transformer based formulation (Section 4 of the main paper) that has separate decoders for the subject and object (namely $f_{\mathbf{s}}$ and $f_{\mathbf{o}}$, MOTIF uses a single network to predict both the subject and object labels.

**Entity Network.** To obtain subject/object labels, for each entity $\mathbf{e}_i$, MOTIF [55] uses a bidirectional LSTM to obtain the set of contextual entity representations $\{\mathbf{c}_i^e\}$,

$$\{\mathbf{c}_i^e\} = \texttt{biLSTM}\left(\{\mathbf{z}_i, \mathbf{l}_i\}\right) \tag{A1}$$

where $\mathbf{c}_i^e$ is the concatenation of the final hidden states of the bidirectional LSTM. A decoder LSTM then sequentially generates labels $\{\widehat{\mathbf{l}}_i^e\}$ for each entity as follows,

$$\mathbf{h}_i = \texttt{LSTM}\left(\mathbf{c}_i^e, \widehat{\mathbf{l}}_{i-1}^e\right)$$
$$\widehat{\mathbf{l}}_i^e = \texttt{argmax}\left(\mathbf{W}_e \mathbf{h}_i\right) \tag{A2}$$

where $\mathbf{W}_e$ is a learned parameter.

**Predicate Network.** Similar to the entity network, the predicate network also uses a bidirectional LSTM to obtain contextual predicate representations $\{\mathbf{c}_i^p\}$ as follows,

$$\{\mathbf{c}_i^p\} = \texttt{biLSTM}\left(\{\mathbf{c}_i^e, \widehat{\mathbf{l}}_i^e\}\right). \tag{A3}$$

For each candidate pair $(\mathbf{e}_i, \mathbf{e}_j)$, the predicate label $\{\widehat{\mathbf{l}}^e_{i,j}\}$ is then obtained as follows,

$$
\begin{aligned}
\mathbf{g}_{i,j} &= \left(\mathbf{W}_h \mathbf{c}^p_i\right) \odot \left(\mathbf{W}_b \mathbf{c}^p_j\right) \odot \mathbf{u}_{i,j} \\
\widehat{\mathbf{l}}^e_{i,j} &= \texttt{argmax}\left(\mathbf{W}_p \mathbf{g}_{i,j}\right)
\end{aligned}
\tag{A4}
$$

where $\mathbf{W}_h, \mathbf{W}_b, \mathbf{W}_p$ are learned parameters, and $\odot$ is the element-wise product.

We refer the reader to [55] for a more detailed explanation of the entity and predicate networks. Our iterative refinement augmented variant of MOTIF defines $T$ entity and predicate networks, denoted by $f^t_{\mathbf{s},\mathbf{o}}$ and $f^t_{\mathbf{p}}$ at layer $t$. Following our formulation described in Section 3 of the main paper, we condition the inputs to the entity $f^t_{\mathbf{s},\mathbf{o}}$ and predicate $f^t_{\mathbf{p}}$ networks appropriately. The two types of conditionings, namely conditioning within a step (CWS) and conditioning across steps (CAS) are implemented as follows,

**Conditioning Within a Step.** Contrary to the transformer based model where we need to explicitly condition with a step (Equation 3 of the main paper), this is already implemented in the default MOTIF [55] architecture. This is evident from Equation A3 wherein the contextual predicate representations $\{\mathbf{c}^p_i\}$ are obtained by using the entity contextual representations $\{\mathbf{c}^e_i\}$. This implicitly defines the following conditioning $\mathbf{e}_i \rightarrow \mathbf{p}_i$.

**Conditioning Across Steps.** In accordance with Equation 2 of the main paper, we want to condition the predictions at layer $t$ on the graph estimate at layer $t-1$. This is achieved by modifying the inputs $\{\mathbf{z}_i\}$ and $\{\mathbf{u}_{i,j}\}$. Specifically, for a particular step $t$, the input to the entity network $\{\mathbf{z}^t_i\}$ is obtained as follows,

$$
\begin{aligned}
\{\widehat{\mathbf{z}}^t_i\} &= \{\mathbf{z}^{t-1}_i\} + \texttt{FFN}\left(\texttt{MultiHead}\left(\{\mathbf{z}^{t-1}_i\}, \{\mathbf{h}^{t-1}_i\}, \{\mathbf{h}^{t-1}_i\}\right)\right) \\
\{\overline{\mathbf{z}}^t_i\} &= \{\widehat{\mathbf{z}}^t_i\} + \texttt{FFN}\left(\texttt{MultiHead}\left(\{\widehat{\mathbf{z}}^t_i\}, \{\mathbf{g}^{t-1}_{i,j}\}, \{\mathbf{g}^{t-1}_{i,j}\}\right)\right) \\
\{\mathbf{z}^t_i\} &= \{\overline{\mathbf{z}}^t_i\} + \texttt{FFN}\left(\texttt{MultiHead}\left(\{\overline{\mathbf{z}}^t_i\}, \{\mathbf{u}^{t-1}_{i,j}\}, \{\mathbf{u}^{t-1}_{i,j}\}\right)\right)
\end{aligned}
\tag{A5}
$$

Similarly, for a particular step $t$, the input to the predicate network $\{\mathbf{u}^t_i\}$ is obtained as follows,

$$
\begin{aligned}
\{\widehat{\mathbf{u}}^t_{i,j}\} &= \{\mathbf{u}^{t-1}_{i,j}\} + \texttt{FFN}\left(\texttt{MultiHead}\left(\{\mathbf{u}^{t-1}_{i,j}\}, \{\mathbf{h}^{t-1}_i\}, \{\mathbf{h}^{t-1}_i\}\right)\right) \\
\{\overline{\mathbf{u}}^t_{i,j}\} &= \{\widehat{\mathbf{u}}^t_{i,j}\} + \texttt{FFN}\left(\texttt{MultiHead}\left(\{\widehat{\mathbf{u}}^t_{i,j}\}, \{\mathbf{g}^{t-1}_{i,j}\}, \{\mathbf{g}^{t-1}_{i,j}\}\right)\right) \\
\{\mathbf{u}^t_{i,j}\} &= \{\overline{\mathbf{u}}^{t-1}_{i,j}\} + \texttt{FFN}\left(\texttt{MultiHead}\left(\{\overline{\mathbf{u}}^{t-1}_{i,j}\}, \{\mathbf{z}^{t-1}_i\}, \{\mathbf{z}^{t-1}_i\}\right)\right)
\end{aligned}
\tag{A6}
$$

Finally, to allow each layer $t$ to refine the estimates of the previous layer $t-1$, the representations $\mathbf{h}^t_i$ and $\mathbf{g}^t_{i,j}$ are modelled as residual connections. Specifically, we modify Equations A2 and A4 as follows,

$$
\begin{aligned}
\mathbf{h}^t_i &= \mathbf{h}^{t-1}_i + \texttt{LSTM}\left(\mathbf{c}^{t,e}_i, \widehat{\mathbf{l}}^{t,e}_{i-1}\right) \\
\mathbf{g}^t_{i,j} &= \mathbf{g}^{t-1}_{i,j} + \left(\mathbf{W}^t_h \mathbf{c}^{t,p}_i\right) \odot \left(\mathbf{W}^t_b \mathbf{c}^{t,p}_j\right) \odot \mathbf{u}^t_{i,j}
\end{aligned}
\tag{A7}
$$

# B   Additional Implementation Details

**Number of Epochs and Model Selection.** The transformer models are trained for 50 epochs on Visual Genome [30] and 18 epochs on Action Genome [25]. The best model is selected by checking the validation set performance in each of the datasets.

**Hardware.** The training is done on 4 Tesla T4 for the transformer models. For the MOTIF augmentation, the training for both the baseline and our variant is done on 4 NVidia A100 GPUs.

**Generating Graph from Triplets.** As our transformer model generates relation triplets, we obtain a graph by applying a non-maximum suppression (NMS) strategy [15, 41] to group the predicted subjects and objects into entity instances. This NMS is applied separately *per class*, and each predicted subject/object is then assigned to a post-NMS bounding box by checking the IoU overlap.

**Data Splits.** For the Visual Genome dataset [30], we follow the widely popular data splits provided by [55], which can be found at the official release repository of [46]:

. For the Action Genome dataset [25], to facilitate fair comparison with the baselines, we contacted the authors of [16] and obtained their data pre-processing code.

The code for our approach is available at github.com/ubc-vision/IterativeSG.

## C  Additional Results

### C.1  Additional Ablations

Table A1: **Refinement Across Steps.** Results are shown for the model trained using $\alpha = 0.14, \beta = 0.75$ on the Visual Genome test set. To measure the scene graph generation performance, we report recall, mean recall, and harmonic recall. To measure the detection performance, we report the box AP, $AP_{50}$, and $AP_{75}$ to analyze the detection performance. Finally, we report the relative model size compared to the final model at each refinement layer.

| t | mR@20/50 | R@20/50 | hR@20/50 | AP | $AP_{50}$ | $AP_{75}$ | Model Size |
|---|---|---|---|---|---|---|---|
| 1 | 9.4 / 13.3 | 18.4 / 23.3 | 12.4 / 16.9 | 11.8 | 24.2 | 9.7 | $0.61\times$ |
| 2 | 10.7 / 14.8 | 19.8 / 24.8 | 13.9 / 18.5 | 13.5 | 26.3 | 11.8 | $0.69\times$ |
| 3 | 11.1 / 14.8 | 20.2 / 25.2 | 14.3 / 18.6 | 14.0 | 26.8 | 12.5 | $0.76\times$ |
| 4 | 11.7 / 15.7 | 20.8 / 26.0 | 15.0 / 19.6 | 14.5 | 27.4 | 13.0 | $0.84\times$ |
| 5 | 11.8 / 15.6 | 21.0 / 26.1 | 15.1 / 19.5 | 14.6 | 27.6 | 13.1 | $0.92\times$ |
| 6 | **11.8 / 15.8** | **21.0 / 26.1** | **15.1 / 19.7** | **14.6** | **27.6** | **13.2** | $1\times$ |

**Refinement Across Steps.**  To further analyze the effectiveness of our refinement procedure, we contrast the quality of scene graphs generated at each refinement step for a particular trained model ($\alpha = 0.14, \beta = 0.75$) in Table A1. From the improvements in recall, mean recall, and harmonic recall at each $t$, it can be reasoned that the quality of the generated graphs steadily improves with each refinement step, wherein the biggest gains are observed in the initial stages. Furthermore, the improvement in the AP numbers indicate that the detection performance improves simultaneously alongside the graph generation performance, highlighting the ability of our approach to jointly reason over subjects, objects, and predicates. Additionally, as each layer $t$ generates a valid scene graph, an added benefit of our proposed formulation is its ability to allow model compression without the need for re-training. One can simply select the first $t$ refinement layers (and discard the rest) depending on the desired accuracy-memory trade-off. For example, from Table A1, selecting the first 4 layers (and discarding the last 2) gives a model that is $0.84$ the size at the expense of a $0.1$ drop in hR@50 and hR@100.

Table A2: **Number of Queries Ablation.** We vary the number of queries used by our transformer decoders and report recall, mean recall, and harmonic recall on the Visual Genome test set. This ablation assume $\alpha = 0.14, \beta = 0.75$.

| # Queries | mR@20/50 | R@20/50 | hR@20/50 |
|---|---|---|---|
| 150 | 11.7 / 15.0 | 21.0 / 25.6 | 15.0 / 18.9 |
| 300 | **11.8 / 15.8** | **21.0** / 26.1 | **15.1 / 19.7** |
| 450 | 11.6 / 15.7 | 20.7 / **26.2** | 14.9 / 19.6 |

**Number of Queries.**  We additionally ablate the number of queries used by the decoders of our transformer model. We report performance on recall, mean recall, and harmonic recall on the Visual Genome [55] dataset in Table A2, assuming $\alpha = 0.14, \beta = 0.75$ for all models. It can be seen that setting the number of queries to be too low or too high is detrimental to performance.

**Zero Shot Recall.** Introduced in [36], zero shot recall (zsR@K) measures recall@K for <subject, object, predicate> that are *absent* from the training set. Therefore, this measure allows analysis of a model performance on unseen relationships. We report zsR@50/100 for certain baselines (including concurrent work in [31]) and our approach in Table A3. It can be seen that our approach in

Table A3: **Zero Shot Recall.** We report zero shot recall for baselines and our approach. The baseline numbers are borrowed from [31].

| # | Model | zR@50/100 |
|---|-------|-----------|
| ① | BGNN [32] | 0.4 / 0.9 |
| ② | DT2-ACBS [11] | 0.3 / 0.5 |
| ③ | VCTree-TDE [46] | 2.6 / 3.2 |
| ④ | SGTR$_{M=3}$ [31] | 2.4 / **5.8** |
| ⑤ | Ours$_{(\alpha=0.14,\beta=0.75)}$ | 2.8 / 3.7 |
| ⑥ | Ours$_{(\alpha=0.14,\beta=0.75),M=3}$ | **3.9** / 5.6 |

⑤ provides the best zsR@50 (0.2 higher when compared to ③), which is a stricter metric. Although SGTR [31] (④) has a higher performance on zsR@100, as described in Section 5.2 of the main paper, it selects top-3 subjects and objects for each predicate, leading to more relationship triplets being generated. Therefore, for a fairer comparison, we evaluate our model in ⑤ using a similar top-$k$ strategy, wherein we select the top-3 predicates for each subject-object pair. This is done by getting the 3 most likely predicate classes from the distribution generated by the predicate decoder. The resulting model (⑥) provides a much greater margin on zsR@50 (1.5 higher than SGTR [31]) while being comparable on zsR@100 (0.2 lower).

**Per Class Recall.** We show the per-class predicate recalls on Visual Genome [30] in Figure A1. Additionally, we contrast the per class recalls between steps $t = 1$ (which corresponds to the model with no refinement) and $t = 6$. It can be seen that with more refinement steps, the model is able to perform considerably better across most classes. A similar observation can be made for Action Genome [25], as shown in Figure A2.

**Per Class AP.** We show the per-class object APs on Visual Genome [30] in Figure A3. Additionally, we contrast the per class APs between steps $t = 1$ (which corresponds to the model with no refinement) and $t = 6$. It can be seen that with more refinement steps, the model is able to perform considerably better across object classes. Note that, for brevity, we only show the top 50 object classes, selected according to the difference between class AP at step $t = 1$ and step $t = 6$. Per-class object APs for the Action Genome [25] dataset is shown in Figure A4.

## C.2 Complementarity to Data Sampling Approaches

In Section 5.2 of the main paper, we briefly highlight that our proposed loss weighting strategy is complementary to existing data sampling approaches by fine tuning our model in ⑳ (Table 1 of the main paper) using BGNN [32]. This fine-tuning is done *only* on the final predicate softmax layer, and all other parameters are frozen. For ease of readability, we copy relevant results from Table 1 of the main paper into Table A4. We show our approach fine-tuned using BGNN [32] in ⑥ (Table A4) provides a considerable improvement over our approach without BGNN (④; 1.3 mR@50 higher with similar hR@50/100 numbers). In comparison to the baseline SGTR [31] with BGNN (③), our model in ⑥ provides better mR@50 (1.3 higher), R@50/100 (2.3/0.7 higher), and hR 50 (1.6 higher). As described in Section 5.2, the poorer performance on mR@100 and hR@100 is largely due to SGTR [31] selecting top-3 subjects and objects for each predicate, leading to more relationship triplets being generated. Therefore, for a fairer comparison, we evaluate our model in ⑥ using a similar top-$k$ strategy, wherein we select the top-3 predicates for each subject-object pair. The resulting model in ⑦ outperforms the baseline in ③ by a significant margin on all metrics, leading to improved recall on head, body, and tail classes. Therefore, our proposed loss weighting scheme can be used in addition to existing data sampling strategies, making it more flexible.

## C.3 Qualitative Results

We provide additional qualitative results for our transformer model at different steps in Figures A5-A14. These highlight steady improvements in the graph quality over refinement steps.

Table A4: **Complementarity to Data Sampling Approaches.** Mean Recall (mR@K), Recall (R@K), and Harmonic Recall (hR@K) shown for the baseline and our approach. **B**=Backbone, **D**=Detector. Baseline results are borrowed from [31]. $M$ indicates the number of top-k links used by the baseline [31].

| B | D | # | Method | mR@50/100 | R@50/100 | hR@50/100 | Head | Body | Tail |
|---|---|---|---|---|---|---|---|---|---|
| ResNet-101 | DETR | ① | SGTR$_{M=1}$ [31] | 12.0 / 14.6 | 25.1 / 26.6 | 16.2 / 18.8 | 27.1 | 17.2 | 6.9 |
| | | ② | SGTR$_{M=3}$ [31] | 12.0 / 15.2 | 24.6 / 28.4 | 16.1 / 19.8 | 28.2 | 18.6 | 7.1 |
| | | ③ | SGTR$_{M=3,\text{BGNN [32]}}$ [31] | 15.8 / 20.1 | 20.6 / 25.0 | 17.9 / 22.3 | 21.7 | 21.6 | 17.1 |
| | | ④ | Ours$_{(\alpha=0.14,\beta=0.75)}$ | 15.8 / 18.2 | 26.1 / 28.7 | 19.7 / 22.3 | 28.2 | 19.4 | 13.8 |
| | | ⑤ | Ours$_{(\alpha=0.14,\beta=0.75),M=3}$ | 19.5 / 23.4 | **30.8 / 35.6** | **23.9** / 28.2 | **32.9** | 28.1 | 15.8 |
| | | ⑥ | Ours$_{(\alpha=0.14,\beta=0.75),\text{BGNN [32]}}$ | 17.1 / 19.2 | 22.9 / 25.7 | 19.6 / 22.0 | 24.4 | 20.2 | 16.4 |
| | | ⑦ | Ours$_{(\alpha=0.14,\beta=0.75),\text{BGNN [32]},M=3}$ | **20.2 / 24.1** | 29.0 / 34.2 | 23.8 / **28.3** | 27.7 | **30.1** | **18.5** |

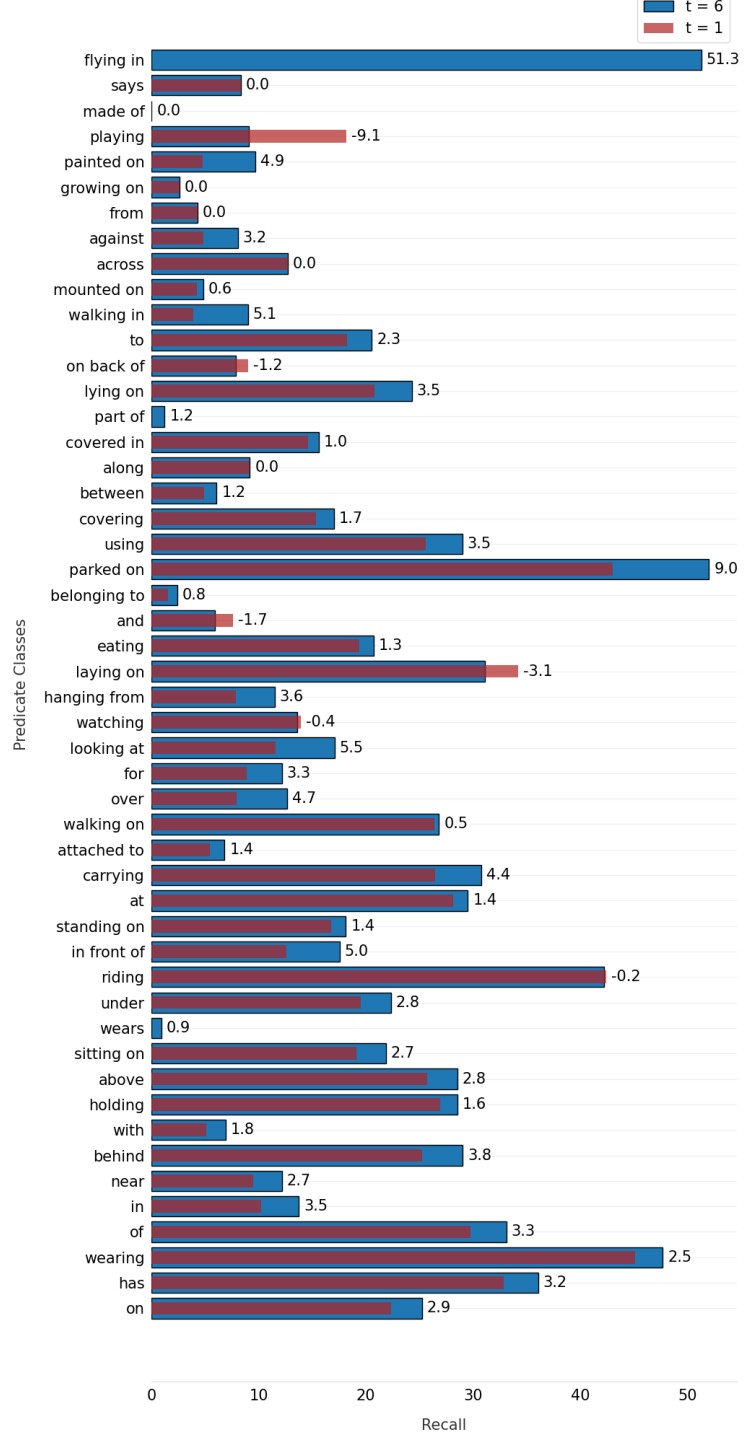

Figure A1: **Per Class Recall on Visual Genome.** Plot shows the recall for the individual predicate classes in Visual Genome [30] for our model trained with $\alpha = 0.14, \beta = 0.75$. The y-axis is sorted in increasing order of predicate frequency in the training set, with the more popular (head) classes at the bottom, and the less popular (tail) classes at the top. We additionally contrast the per class recalls between steps $t = 1$ (indicating no refinement; in red) and $t = 6$ (in blue). The number next to each bar indicates the difference between recall at $t = 6$ and $t = 1$ for a particular predicate class.

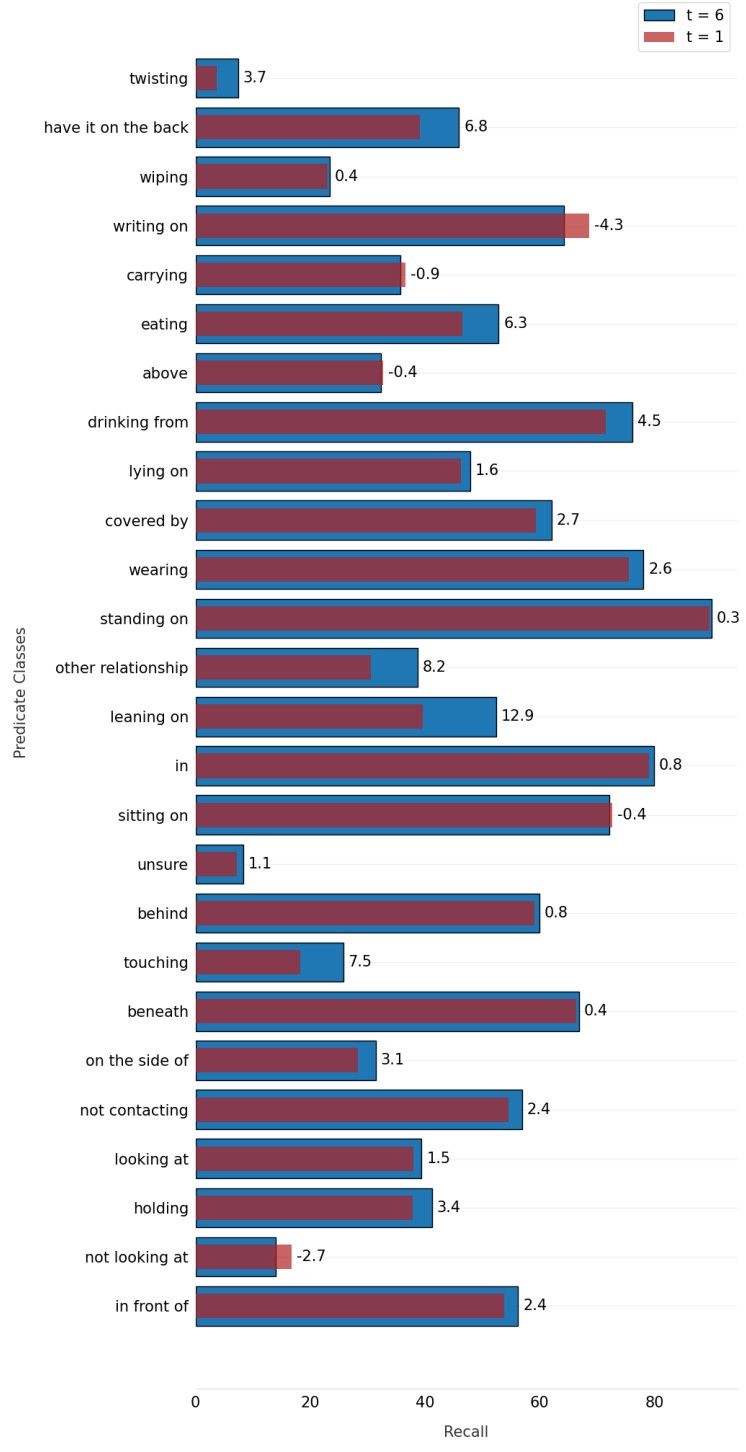

Figure A2: **Per Class Recall on Action Genome.** Plot shows the recall for the individual predicate classes in Action Genome [25] for our model trained with $\alpha = 0.14, \beta = 0.75$. The y-axis is sorted in increasing order of predicate frequency in the training set, with the more popular (head) classes at the bottom, and the less popular (tail) classes at the top. We additionally contrast the per class recalls between steps $t = 1$ (indicating no refinement; in red) and $t = 6$ (in blue). The number next to each bar indicates the difference between recall at $t = 6$ and $t = 1$ for a particular predicate class.

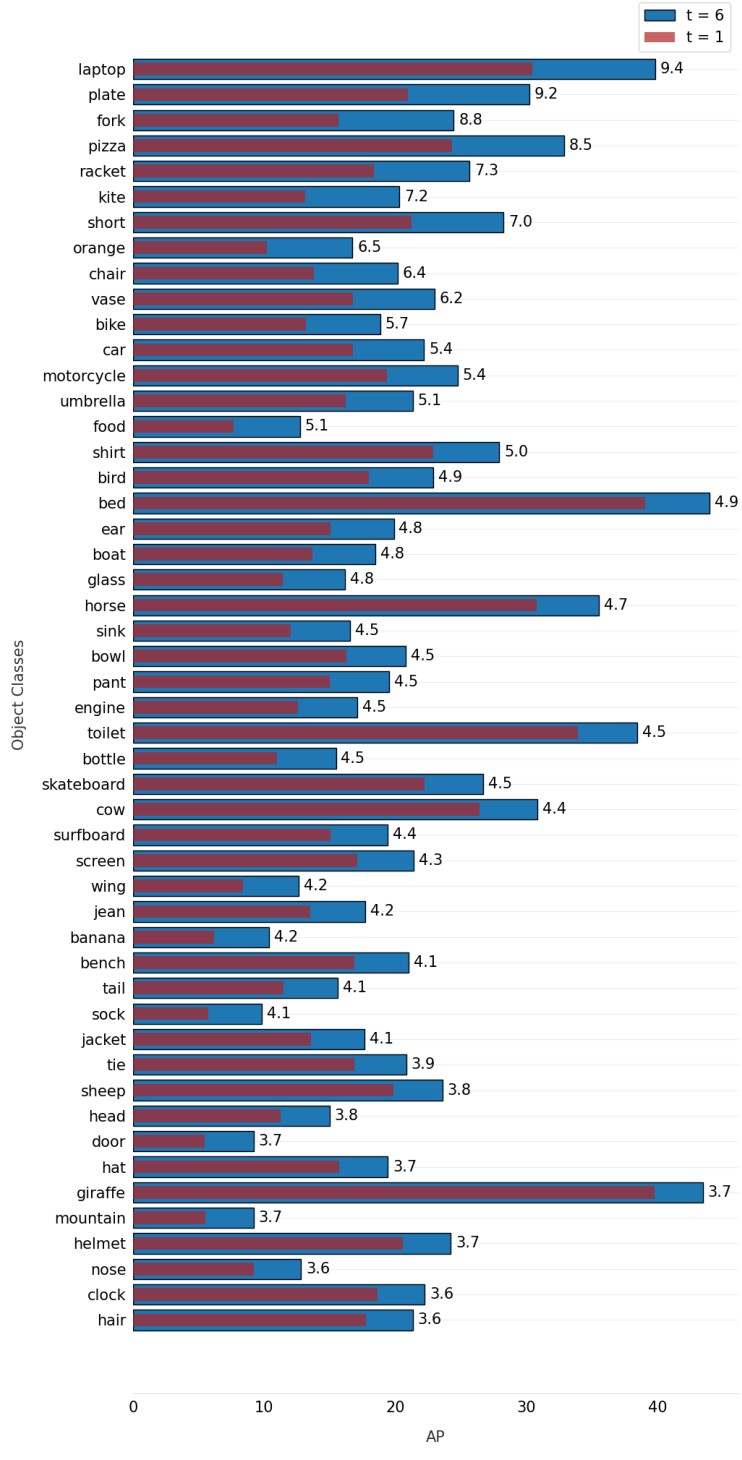

Figure A3: **Per Class AP on Visual Genome.** Plot shows the AP for the top 50 individual object classes in Visual Genome [30] (out of 150 total object classes) for our model trained with $\alpha = 0.14, \beta = 0.75$. We contrast the per class APs between steps $t = 1$ (indicating no refinement; in red) and $t = 6$ (in blue). The number next to each bar indicates the difference between AP values at $t = 6$ and $t = 1$ for a particular object class. The y-axis is sorted by this difference value.

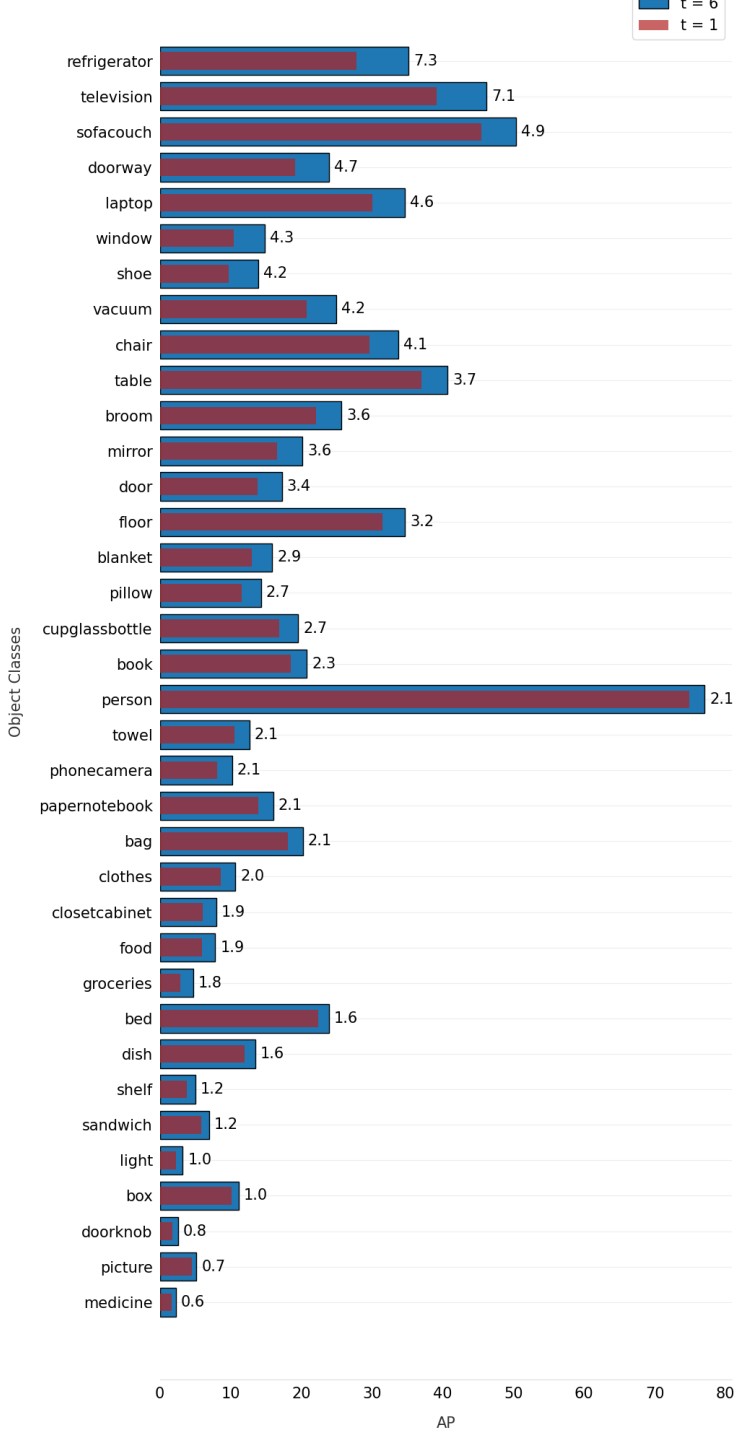

Figure A4: **Per Class AP on Action Genome.** Plot shows the AP for the individual object classes in Action Genome [25] for our model trained with $\alpha = 0.14, \beta = 0.75$. We contrast the per class APs between steps $t = 1$ (indicating no refinement; in red) and $t = 6$ (in blue). The number next to each bar indicates the difference between AP values at $t = 6$ and $t = 1$ for a particular object class. The y-axis is sorted by this difference value.

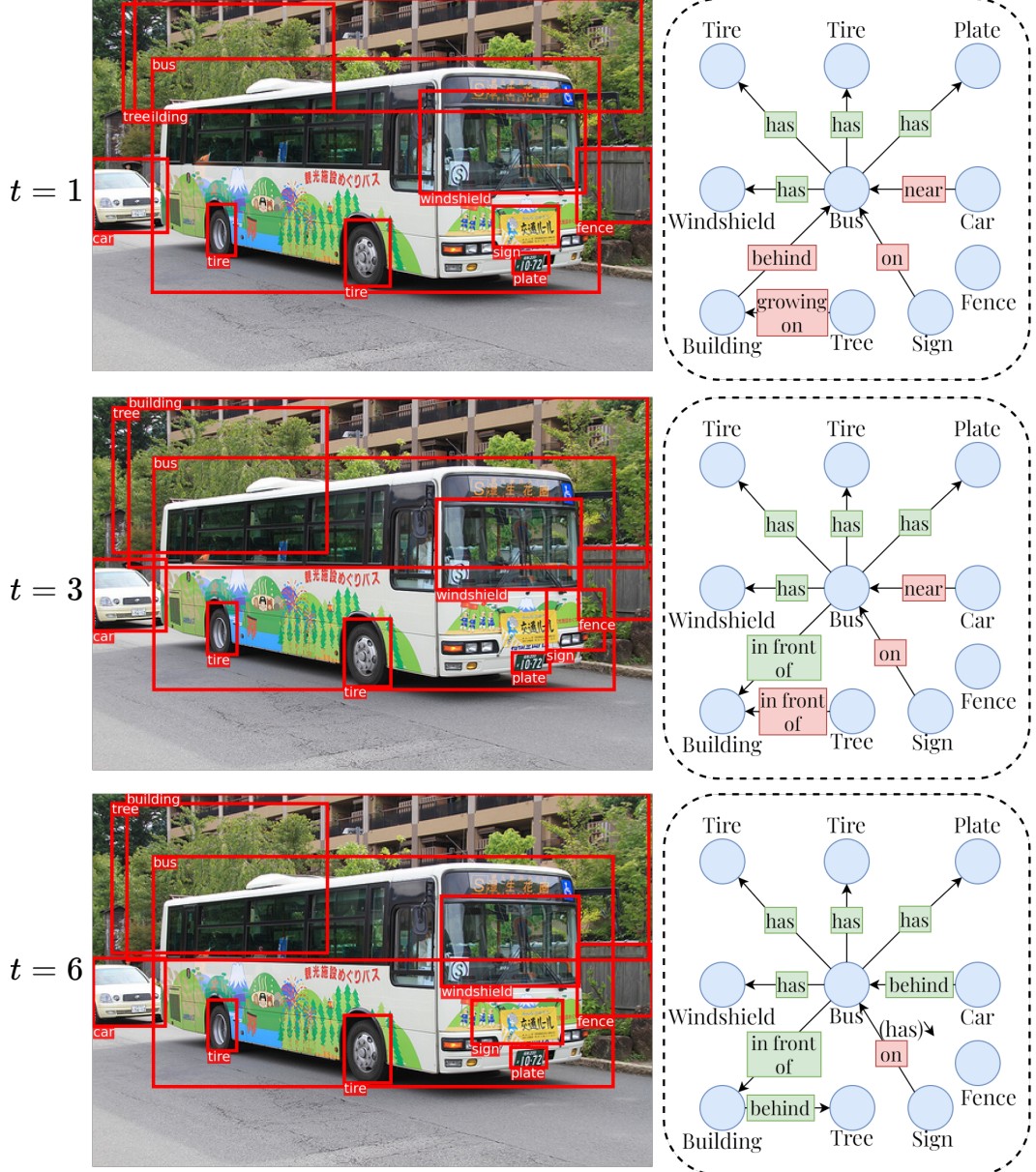

Figure A5: **Qualitative Results.** Graph estimates for different refinement steps ($t = 0, 3,$ and 6 are shown. Colors red and green indicate incorrect and correct predictions respectively. For the incorrect predictions at $t = 6$, we additionally mention the correct predicate label in parenthesis next to it, and also show direction of the relation.

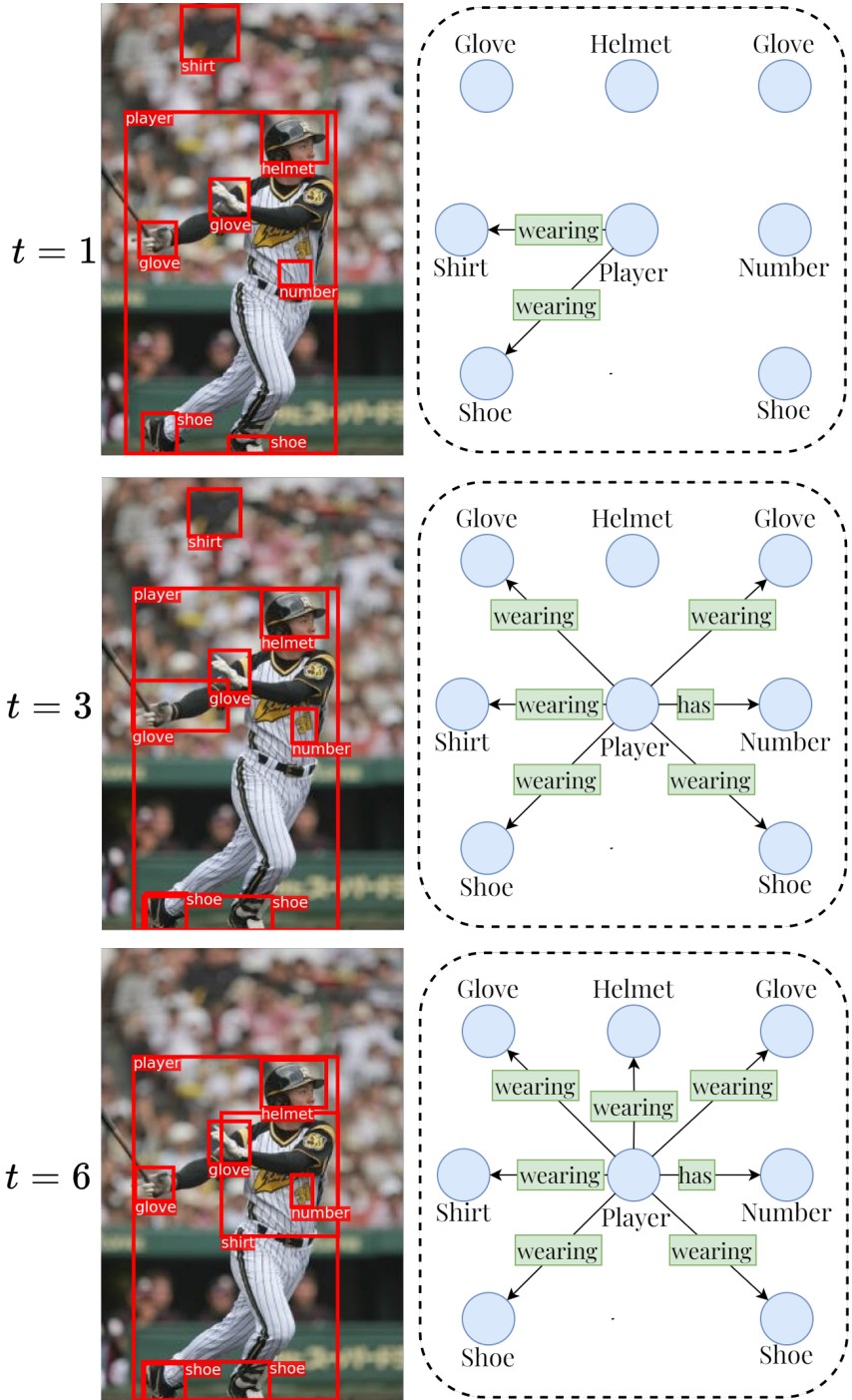

Figure A6: **Qualitative Results.** Graph estimates for different refinement steps ($t = 0, 3$, and 6 are shown. Colors red and green indicate incorrect and correct predictions respectively. For the incorrect predictions at $t = 6$, we additionally mention the correct predicate label in parenthesis next to it, and also show direction of the relation.

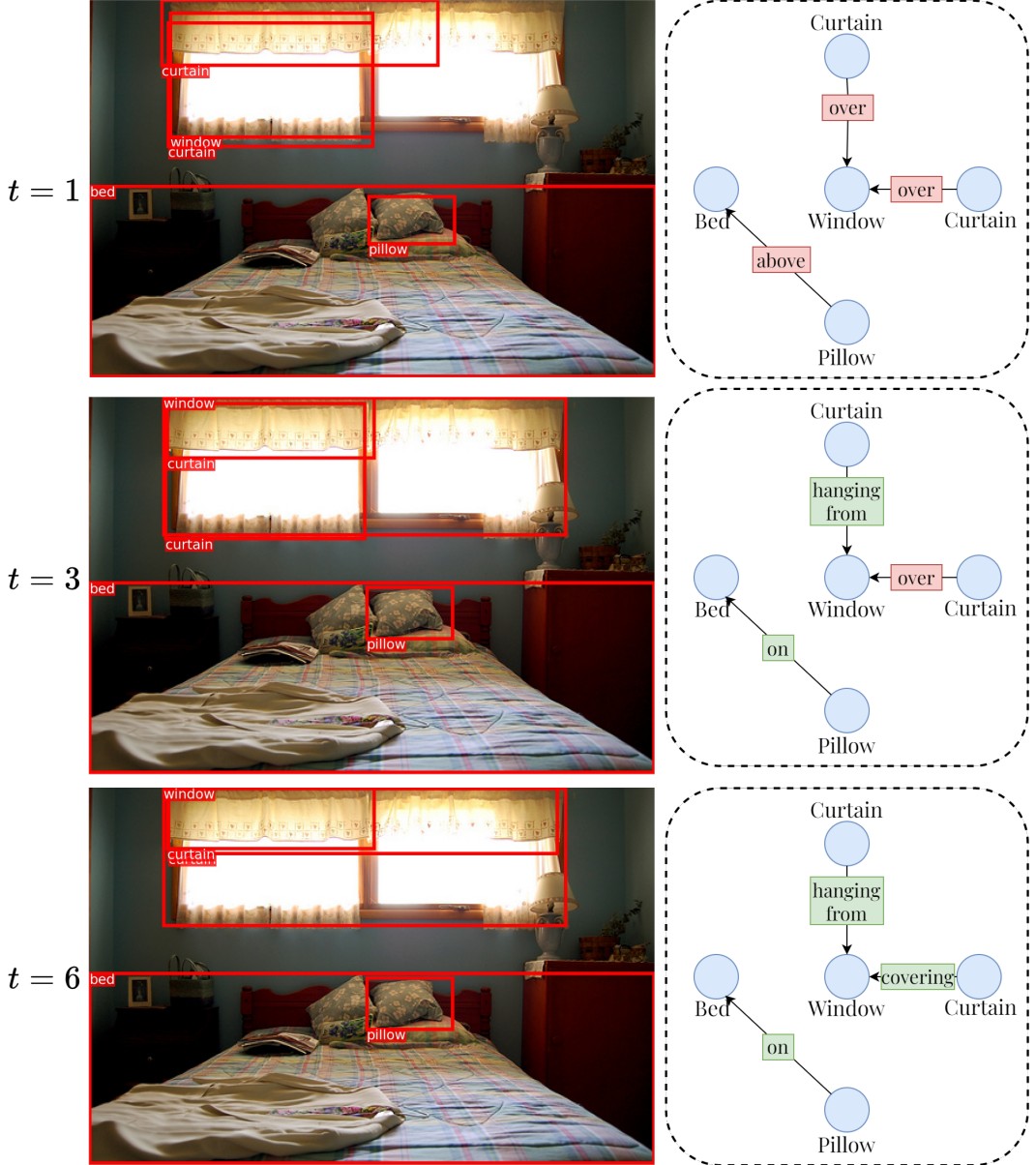

Figure A7: **Qualitative Results.** Graph estimates for different refinement steps ($t = 0, 3$, and6 are shown. Colors red and green indicate incorrect and correct predictions respectively. For the incorrect predictions at $t = 6$, we additionally mention the correct predicate label in parenthesis next to it, and also show direction of the relation.

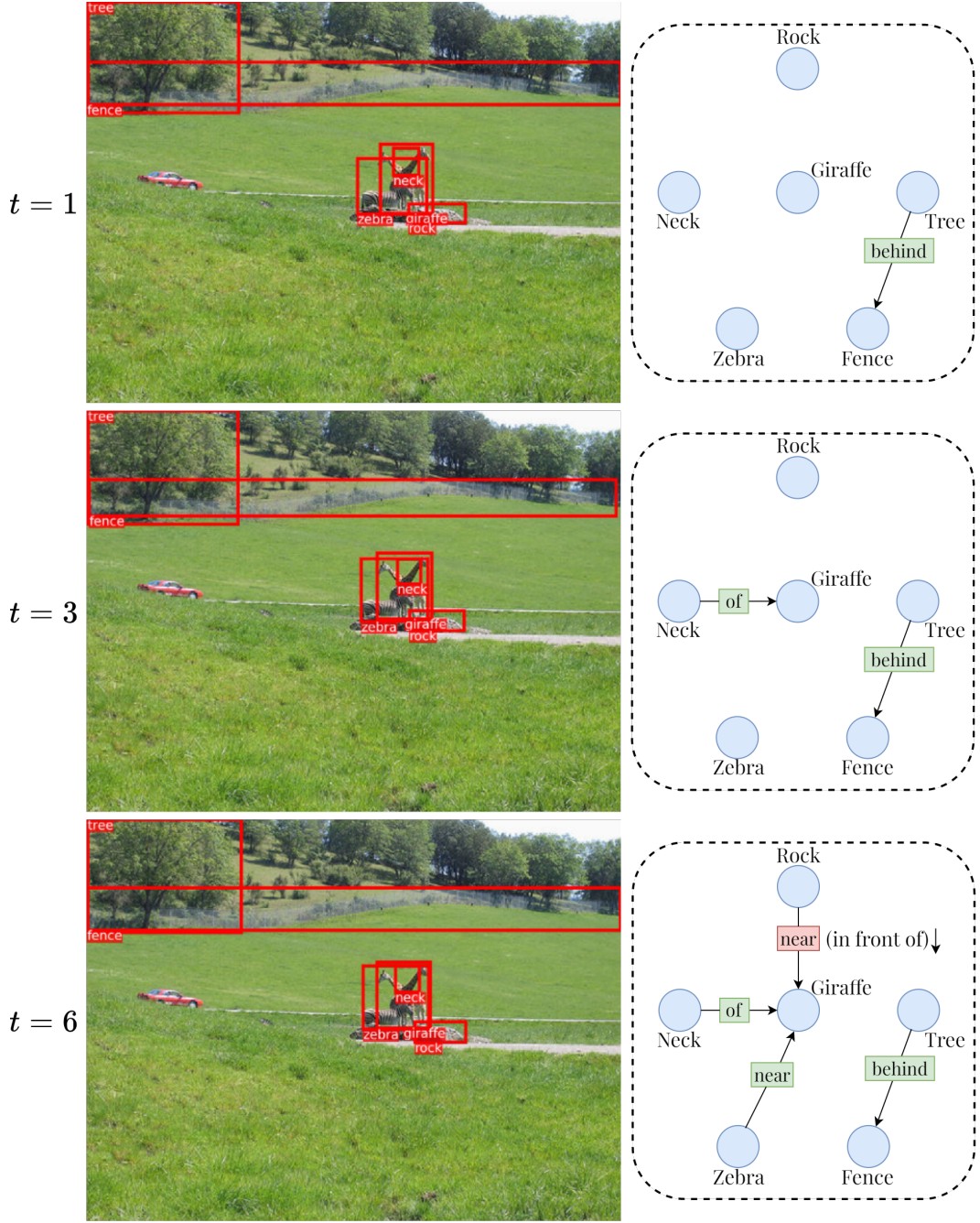

Figure A8: **Qualitative Results.** Graph estimates for different refinement steps ($t = 0, 3$, and6 are shown. Colors red and green indicate incorrect and correct predictions respectively. For the incorrect predictions at $t = 6$, we additionally mention the correct predicate label in parenthesis next to it, and also show direction of the relation.

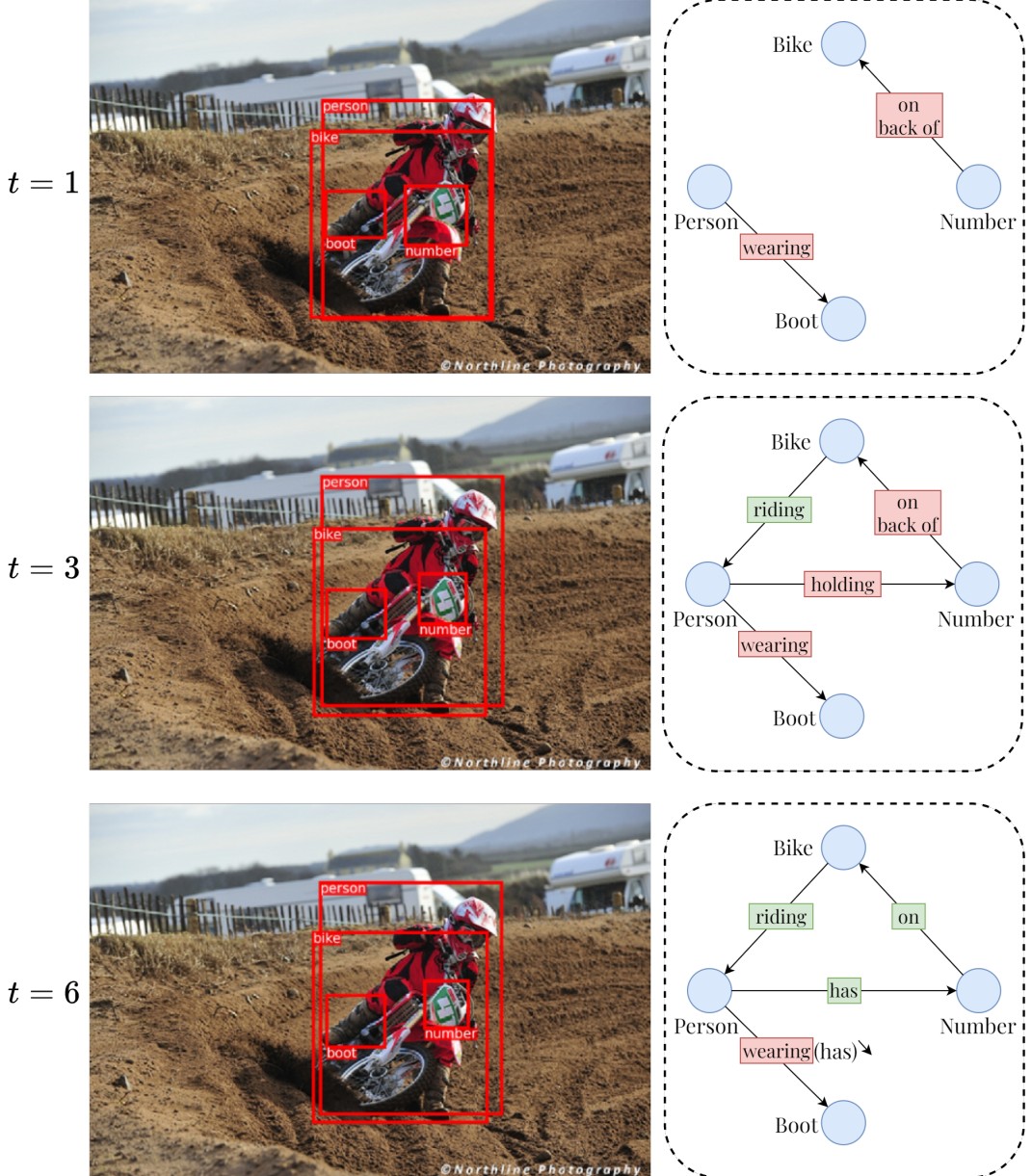

Figure A9: **Qualitative Results.** Graph estimates for different refinement steps ($t = 0, 3$, and6 are shown. Colors red and green indicate incorrect and correct predictions respectively. For the incorrect predictions at $t = 6$, we additionally mention the correct predicate label in parenthesis next to it, and also show direction of the relation.

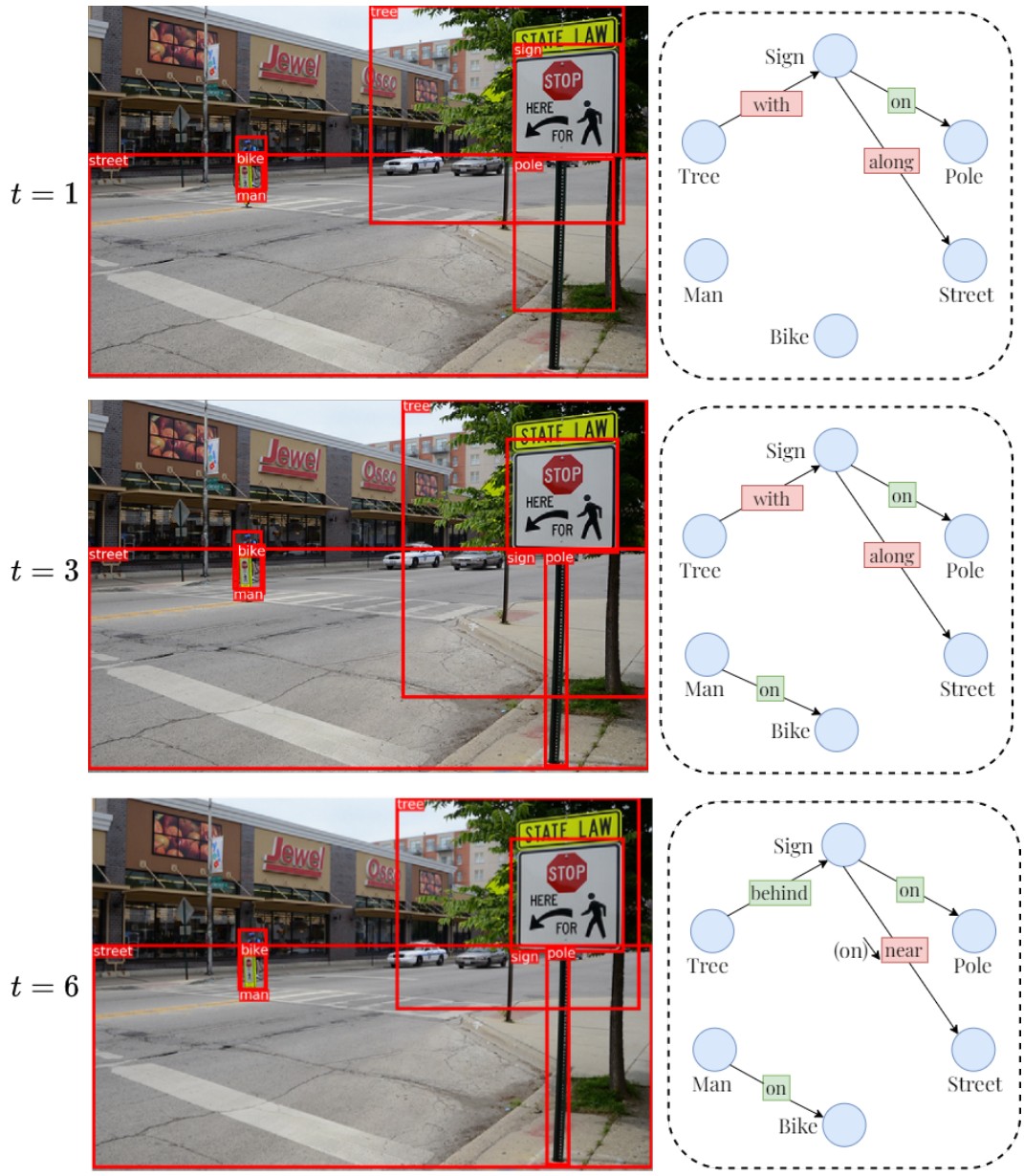

Figure A10: **Qualitative Results.** Graph estimates for different refinement steps ($t = 0, 3$, and 6 are shown. Colors red and green indicate incorrect and correct predictions respectively. For the incorrect predictions at $t = 6$, we additionally mention the correct predicate label in parenthesis next to it, and also show direction of the relation.

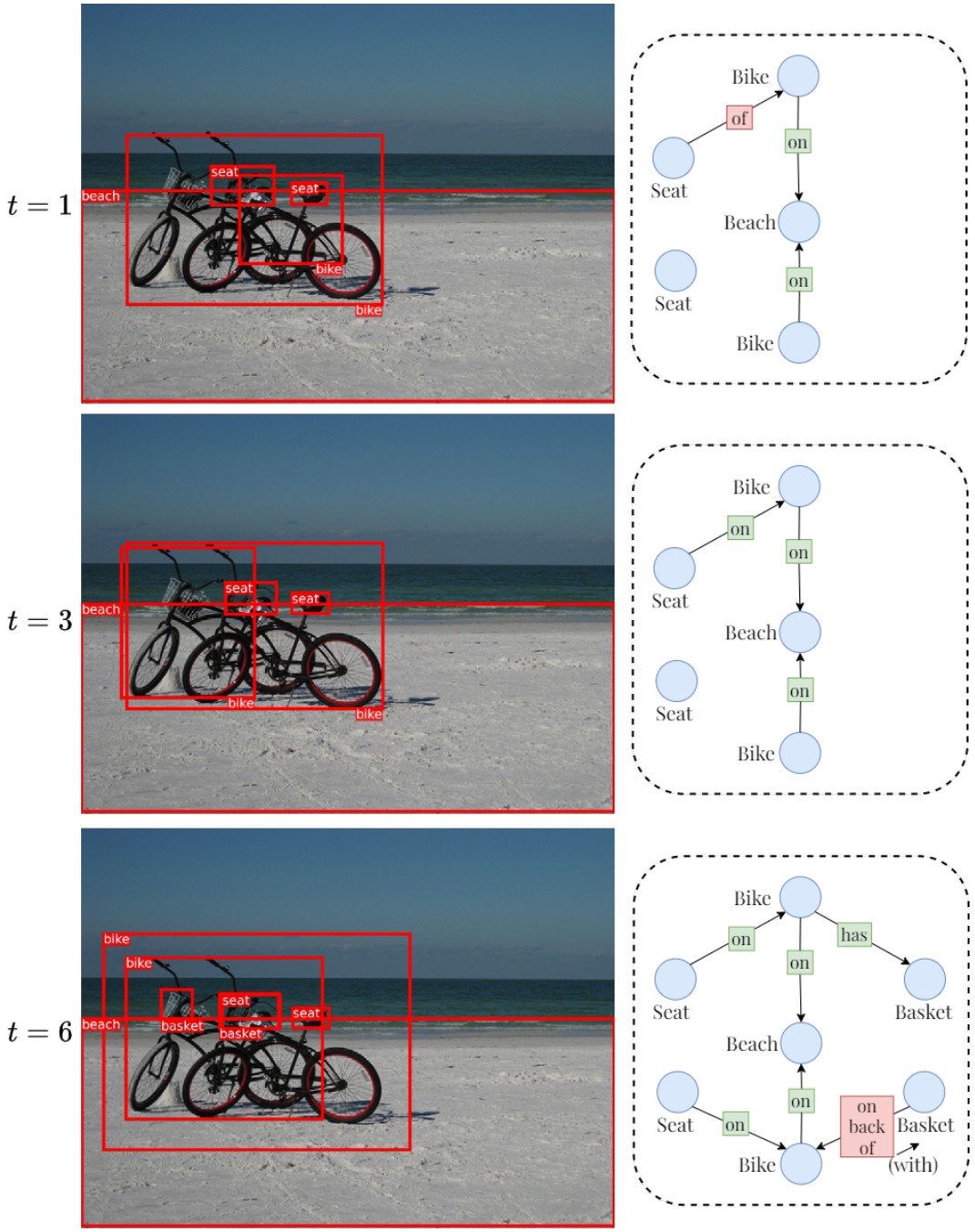

Figure A11: **Qualitative Results.** Graph estimates for different refinement steps ($t = 0, 3$, and $6$ are shown. Colors red and green indicate incorrect and correct predictions respectively. For the incorrect predictions at $t = 6$, we additionally mention the correct predicate label in parenthesis next to it, and also show direction of the relation.

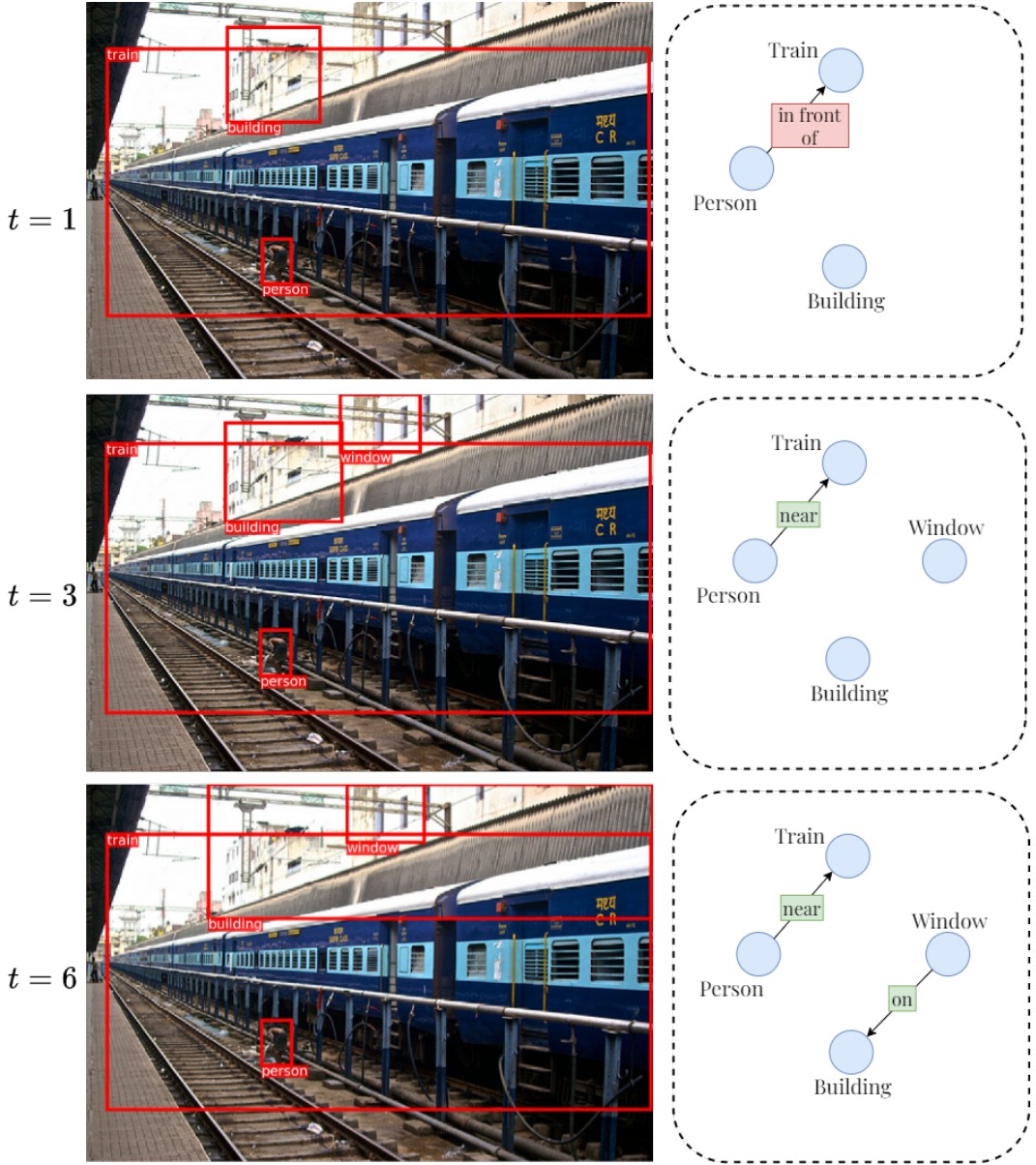

Figure A12: **Qualitative Results.** Graph estimates for different refinement steps ($t = 0, 3,$ and6 are shown. Colors red and green indicate incorrect and correct predictions respectively. For the incorrect predictions at $t = 6$, we additionally mention the correct predicate label in parenthesis next to it, and also show direction of the relation.

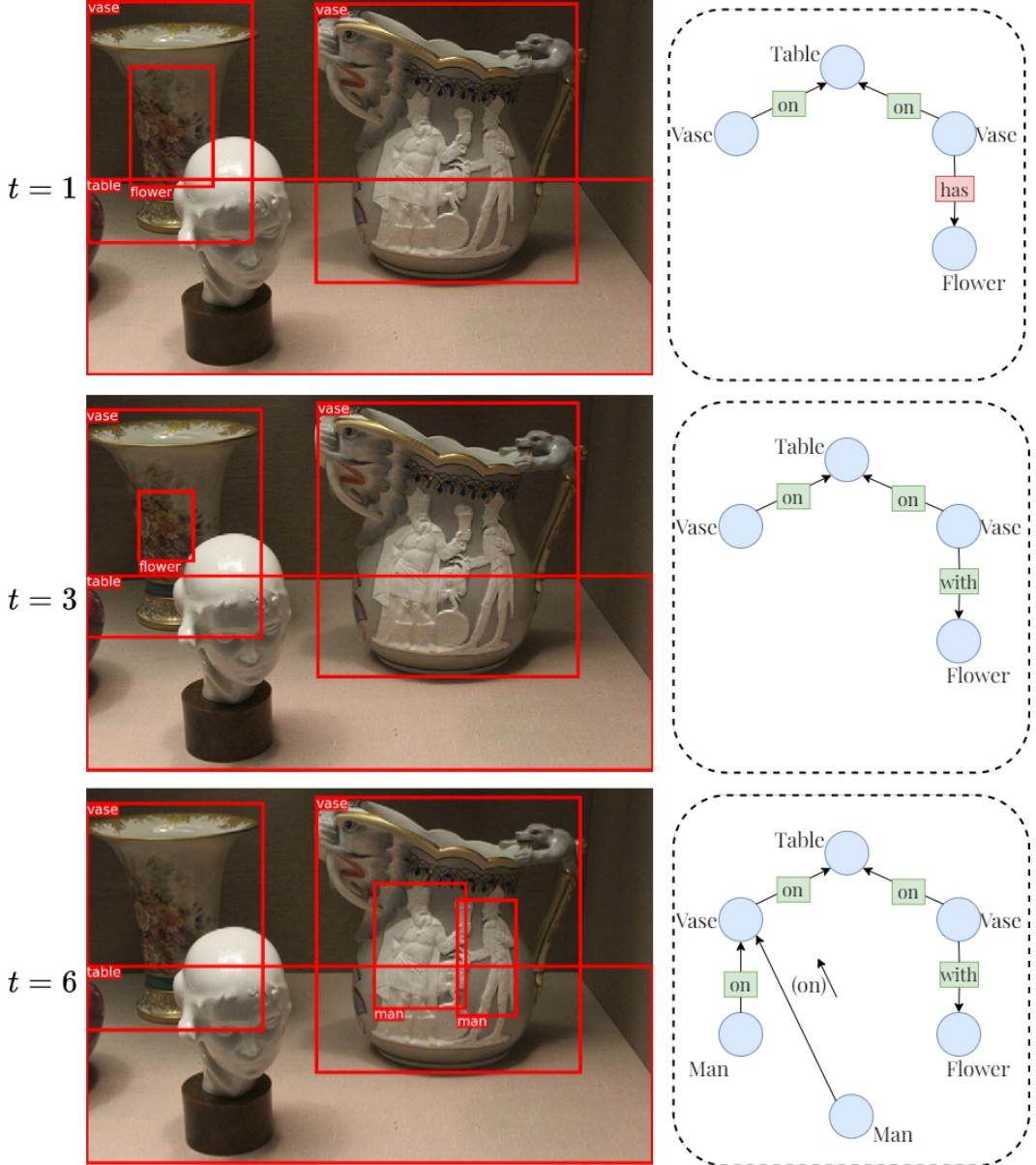

Figure A13: **Qualitative Results.** Graph estimates for different refinement steps ($t = 0, 3$, and 6 are shown. Colors red and green indicate incorrect and correct predictions respectively. For the incorrect predictions at $t = 6$, we additionally mention the correct predicate label in parenthesis next to it, and also show direction of the relation.

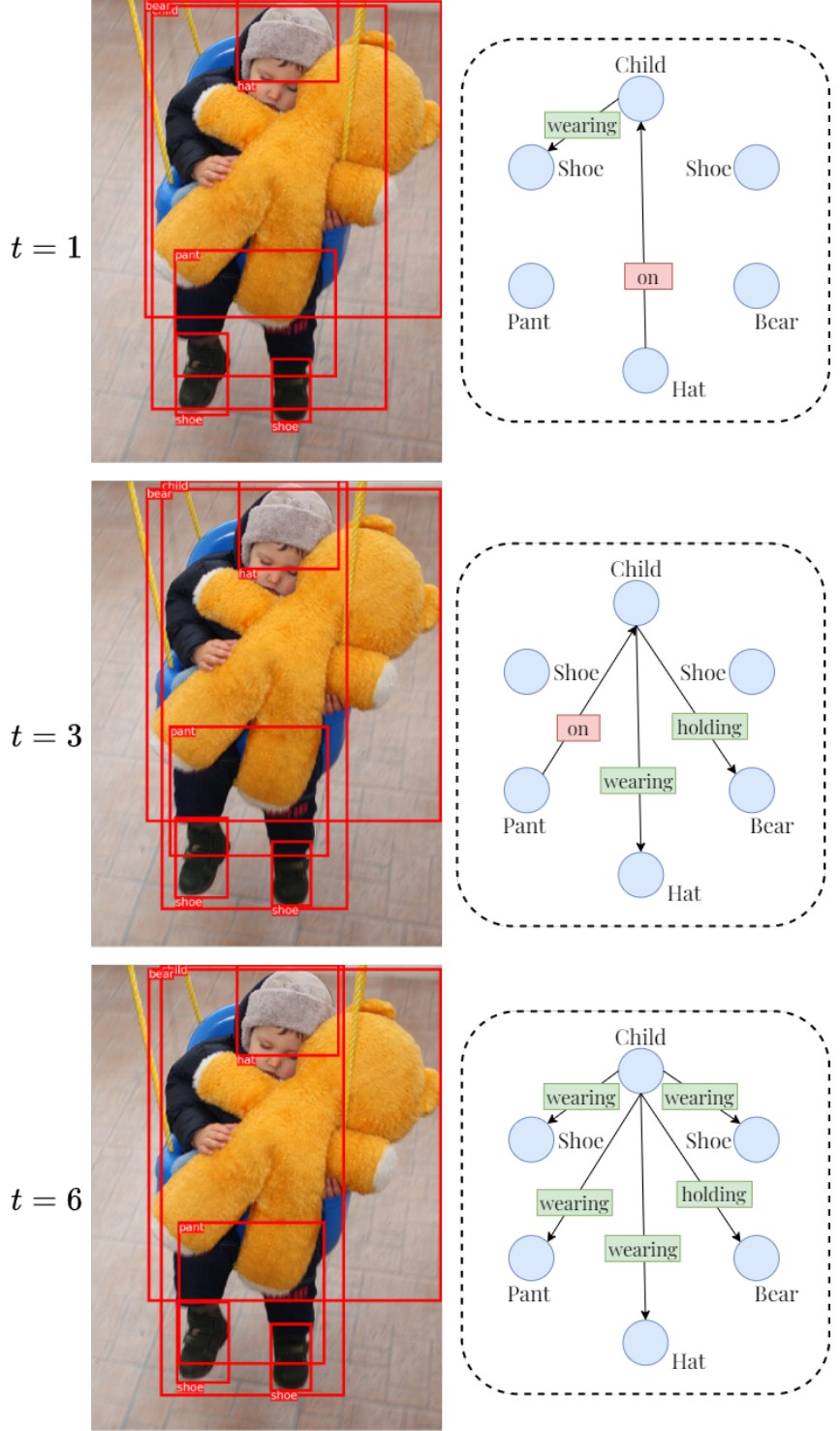

Figure A14: **Qualitative Results.** Graph estimates for different refinement steps ($t = 0, 3$, and 6 are shown. Colors red and green indicate incorrect and correct predictions respectively. For the incorrect predictions at $t = 6$, we additionally mention the correct predicate label in parenthesis next to it, and also show direction of the relation.