# OpenReview forum: "Iterative Scene Graph Generation"
_NeurIPS.cc/2022/Conference — NeurIPS 2022 Accept_

### Official Review · Reviewer_5s37 · 2022-07-11

**Rating:** 4
**Confidence:** 4
**Soundness:** 3 good
**Presentation:** 2 fair
**Contribution:** 2 fair

**Summary:**

The paper proposes dynamic conditioning using Markov Random Filed for the Scene Graph generation. It implemented transformer based iterative refinement procedure . Conditional quaries has been used during the refinement and to reduce the exponential search space. Three separate multi layer decoder has been used for relation triplet and perform joint reasoning in a single stage architecture. It achieved sota performance on VG and AG dataset.

**Questions:**

1. How you have incorporated the method in video, using some tracking? please mention
2. Please provide clear motivation for three decoders and why it is necessary despite their complexity



**Limitations:**

yes.

**Strengths And Weaknesses:**

Strength:
1. The paper proposes a single stage Scene Graph generation approach, with novel transformer based formulation which include separate decoder for relational triplet, and conditional query.
2. The paper show that their results improved VG and AG sota.
3.  Tested the approach on both video and images.

Weakness :
Depite their strength this paper lack few fundamental point :
1. IMHO this paper lacks a strong motivation, author should clearly mention on the introduction why the conditional query is needed and how it can be incorporated to architecture and why they need thee decoder in a one stage..the story should be connected ...
2. What is the initial query, is it same like DETR or how do u process or the initial query are only subject queries ? (sec 4.2)
3. It has three separate layer of decoder and we know that scene graph has exponential complexity...author should analyiz dat

Acknowledgement :
It would be better if you acknowledge  other single stage paper, like Relationformer - Shit at el, RelTr - Cong at el.. and other earlier  transformer based paper like Seq2Seq- Lu at el, Relation Transformer Network - Koner at el

Suggention :
The paper has technical novelty, please write in a way such that it would be easy to follow and understand your contribution

---

> ### Author Response · Authors · 2022-07-29
> **Response to Weaknesses and Questions**
>
> We appreciate the feedback from the reviewer, and clarify the issues and misunderstandings below. We hope that the reviewer will carefully consider them.
>
> $\textbf{Weaknesses}$
>
> $\textbf{1. Design choices - three decoders and conditional queries.}$
>
> Our work proposes a model agnostic iterative refinement framework for scene graph generation (Equation 2). We implement this using a transformer based architecture, wherein each component of factorized distribution in Equation 2 is realized using a decoder. The three separate decoders are merely a tool to accomplish the aforementioned factorization, and predict different components of a relationship triplet. As described in Section 4.2 (lines 196 - 205), the factorization in Equation 2 dictates that each decoder needs to be conditioned in two ways. The conditional queries are used to implement one of those conditionings (over the previous graph estimate), and is a property of the transformer architecture, and not the proposed framework itself. For example, when augmenting our approach to MOTIF [53], we do not have any conditional queries.
>
> In other words, our introduction focuses on the core idea of the need to condition in certain ways, which we believe we motivate well. The “conditional query” and “three decoders” are a specific, but natural, instantiation of this core idea, but by no means the only one. Since this instantiation is not critical to our overarching premise, we spend less time motivating these aspects of design directly.
>
> $\textbf{2. Initial queries.}$
>
> Similar to our approach, DETR [2] also uses queries and positional encodings (also referred to as object queries in [2]). Identical to DETR, our approach initializes the queries at the first step for each decoder to zero (i.e. $ \mathbf{q^1_s} = \mathbf{q^1_o} = \mathbf{q^1_p} = \mathbf{0}$ ). After the generation of a scene graph at the first step, these queries are updated in accordance with Equation 4, and used as input to the second step. This process is repeated $t$ times.
>
> $\textbf{3. The proposed approach does not have an exponential complexity.}$
>
> Although the space of all possible relation triplets is quite large, our model does not evaluate every permutation. The three decoders are not independent of each other, and each predicts a specific component of the scene graph triplet, namely the subject, object, and the predicate. Therefore, the number of triplets our proposed transformer based model generates is equal to the number of queries used (and not exponential).
>
> $\textbf{4. Citing related work.}$
>
> We appreciate pointers to Relationformer (Shit et al.) and RelTr (Cong et al.); they are relevant and we will cite them appropriately. That said, we want to highlight that, to the best of our knowledge, they have only appeared on ArXiv recently and are still formally unpublished. Therefore, in accordance with the NeurIPS guidelines, they should not be considered prior work for the purposes of this submission.
>
> $\textbf{Questions}$
>
> $\textbf{1. Evaluation on videos.}$
>
> As described in Section 5 (lines 255 - 259), we only use annotated frames in the Action Genome dataset [42]. Therefore, we do not evaluate our method on videos per se, rather we treat frames from video as “images” and work with those. Note that this is consistent with the listed Baselines.
>
> $\textbf{2. The need for three decoders.}$
>
> The need for three implicitly coupled decoders arises from our desire to induce conditional independence and structure among the predicted elements of the triplet. The alternatives would be to (1) have independent predictions for objects and predicates that need to be assembled into triplets (this is the approach taken by SGTR [30]) to which we compare in Table 1, or (2) to have a single decoder that would predict a full triplet at once. Note that in the latter case a single decoder would be required to predict both objects, their boxes and predicate labels all in one go. In our preliminary experiments, at the beginning of this project, this proved to be difficult and produced results that weren’t at all competitive with state-of-the-art.

---

> > ### Comment · Reviewer_5s37 · 2022-08-09
> > **Response to Rebuttal**
> >
> > I have read the author response and the design choice or technical novelty may not be good enough.I would advise the author to make the motivation and writing clearer and easier to understand. The result and ablation should be backed by strong performance. And not including RelTr or Relationformer doesn't impact the rating, just good to have.

---

> > > ### Comment · Reviewer_h8U6 · 2022-08-09
> > > **Clarification?**
> > >
> > > @Review5s37: Could you be a bit more specific about this? Specifically, why did the rebuttal fail to address your initial concerns?

---

> > > ### Author Response · Authors · 2022-08-10
> > > **Response to Comment**
> > >
> > > Thank you for reading our rebuttal.
> > >
> > > Overall it is difficult to objectively argue about novelty. However, we would like to highlight that, to the best of our knowledge, the high-level idea of iterative scene graph generation is novel and has not appeared elsewhere. The unique benefit of such formulation is its ability to forego fixed factorization common to all prior methods. This proposed formulation is both general and effective. At a technical level, formulating an instance of such a model using a transformer-based architecture, required us to introduce a novel loss, a synchronized three-stream decoder (for subjects, objects and predicates) and an innovative conditioning scheme across these decoders. In addition, we also study and address the long-tail nature of the scene graph predicate classes by introducing tunable loss weighting that can be adjusted based on demands of the final down-the-stream task.
> > >
> > > We would additionally like to highlight that our results and ablations are indeed backed by strong performance. Specifically, our improvement over the latest state-of-the-art (concurrent work) of SGTR [30] are **very significant** (comparing #16 and #22 in Table 1; our approach is **3.7** higher on mR@50 and **10.2** higher on R@50). In terms of ablations, reported in Table 2, our improvement over the vanilla model (containing the same number of parameters) that does not contain our loss formulation or other components of the proposed model is **9.5** on mR@20 and **17.4** on R@20.

---

### Official Review · Reviewer_h8U6 · 2022-07-11

**Rating:** 6
**Confidence:** 3
**Soundness:** 3 good
**Presentation:** 3 good
**Contribution:** 2 fair

**Summary:**

This paper proposes an end-to-end paradigm to predict scene graphs from image inputs. The key observation of this paper is that assuming a fixed factorization of subject-object-predicate of predicting relationships can be detrimental, as errors can accumulate in this on direction flow of this information. To alleviate this problem, the authors propose an iterative refinement process, where, although the factorization is still the same within a refinement step, information from previous steps can be utilized by later steps, thus allows information to flow in all directions. A joint matching loss is proposed, using the same matching across all steps, to stabilize this refinement procedure. Comprehensive evaluations show that the proposed method achieves good performances and outperforms many prior works.

**Questions:**

As mentioned earlier, my main issue is with the (relatively) lack of analysis of the concrete benefits of the core idea of the paper. More evidence, in addition to "better numbers overall", that it is solving the issues with fixed factorizations will sway me towards accepting this paper.

Some additional comments:
- Might help to highlight the best/top performing entries in the larger tables. Very hard to parse the table as for now.
- Would be interesting to visualize the qualitative behavior of the model without the joint matching loss. I suppose the model is not really doing refinement without that?
- Following previous point: ablations of model with CAS and CWS, but without JL, would be nice to have.

**Limitations:**

Limitations are discussed but those are not limitations specific to the central idea of this paper. More discussions on how to improvement the iterative refinement procedure would be more desirable here.

The authors are right in pointing out that the proposed work takes the right step towards alleviating some of the negative social impacts associated with the problem studied here.


**Strengths And Weaknesses:**

### Strengths
+ The idea of using an iterative refinement procedure to alleviate the issue with a fixed factorization is novel, makes sense intuitively, and seems to work on well in practice.
+ Careful design choices for the aforementioned refinement procedure, including appropriate modification to the inputs of the architecture used (transformers), and a matching loss utilizing the same matching across all steps to ensure the stability of the iterative refinement process.
+ Comprehensive evaluation and good empirical results, with good discussion of some of the nuances over prior works.

### Weaknesses
- Overall, I think there lacks enough evidence supporting the central claim of the paper: that this iterative refinement process is important and avoids the problem with a fixed factorization:
    * It is unclear how the qualitative examples are chosen, and overall too little qualitative examples are shown to support claims that this process fixes the problem with a fixed factorization. More, randomly drawn, examples shown the same qualitative behavior would greatly increase my confidence in the usefulness of this process. I would also expect some results to demonstrate how the proposed method solves issues discussed in the paper e.g. "where an object entailed in interaction is small and not discernible on its own" (L7).
    * Quantitative analysis over the output of each of the refinement step would help to show how the refinement process is actually improving the results. Ablations over the number of steps would also help here. I get that this is somewhat done in Supplementary Table A1, but that can also be due to the models having more parameters overall. I am looking more more fine grained analysis on what specific do the refinement process help the most in improving e.g. if the issue quoted above (L7) is addressed by the proposed method, that can probably be revealed by analysis showing what categories are improved the most over the steps. This currently does not appear to be the case to me since both recall and mean recall follows the same trend in table A1.
    * The fact that removing both CAS and CWS results in minimal performance drop seems to indicate that there isn't much actual ``refinement" going on.
 - The claim that the proposed method can be used to improve other methods is not that well supported: although the performance is marginally better, it is unclear whether it's simply due to the modified model having more parameters or not.
- Originality of the paper is pretty limited apartment from this (still questionable to me) procedure of refinement. The core of the method is still a decoder in the style if [2] and this has been attempted already in earlier literatures e.g. [30].
- In general, I am not a fan of going into very fine details to show that a method outperforms prior/concurrent works e.g. 14-16, 21-22 in Table 1. Showing that the relative advantages of methods have to come down to such details, to me, is rather unnecessary. A good idea should be able to show its value with more specific evaluation methods / examples in addition to a holistic number about overall performance (see points above).

###Post-rebuttal Comment

I thank the authors for the clarifications regarding my concerns. It seems that I initially missed some evaluations, which did provide decent amount of evidence that there is actually a "refinement" process, and gives some level of intuition behind how this process works. Subsequently, I am raising my score from a 4 to a 6. I would encourage the authors to add more discussions around these points in the main paper, as well as providing more qualitative examples in the supplementary.

---

> ### Author Response · Authors · 2022-07-29
> **Response to Weaknesses and Questions.**
>
> We appreciate the feedback from the reviewer, and clarify the issues and misunderstandings below. We hope that the reviewer will carefully consider them.
>
> $\textbf{1. Qualitative examples and performance on smaller objects. }$
>
> The qualitative examples shown in Figure 3 and supplementary Figures A3 - A7 were not chosen with any particular care or criteria. The ability of our approach to better detect smaller objects (line 7) is apparent in Figure 3, where the umbrella on the right is incorrectly localized to a much larger bounding box without any refinement ($t = 1$). As the refinement progresses, the bounding box gets much tighter and more accurate. We will add additional qualitative examples to the supplemental upon revision.
>
> Although you correctly point out that the recall and mean recall follow a similar trend in supplementary table A1, we believe this does not necessarily imply that all categories are improved by the same amount. This is evident from the per class predicate numbers in supplementary Figure A1, where the impact of certain predicates like “$\texttt{flying in}$” is larger than predicates like “$\texttt{says}$“ on the recall and mean recall metrics. To analyze this further, we similarly look at the per class object detection improvements from $t = 1$ (no refinement) to $t=6$. We find that categories like $\texttt{plate}$, $\texttt{fork}$, $\texttt{kite}$, and $\texttt{orange}$, that generally tend to have smaller bounding boxes, have the highest improvements in detection performance ($9.2$, $8.8$, $7.2$, $6.5$ better on AP respectively). This further supports our claim in line 7, and we will add numbers for remaining classes in the supplementary upon revision.
>
> $\textbf{2. Impact of Joint Loss, CAS and CWS. }$
>
> As you correctly point out in your additional comments, the joint matching loss is an important component in the refinement procedure. More specifically, the joint loss induces a strong implicit conditioning by using the same assignment at each refinement step (see Section 4.3). Therefore, even without CAS or CWS, the joint loss by itself enables refinement, which is evident from Table 2 (#3). CAS further builds on the joint loss, and allows for a more structured pathway to leverage any information that the implicit conditioning is unable to capture. This leads to an improvement of around 0.5 on mR@20 and 0.8 on R@20, which is significant in the scene graph literature. For example, in [43] the maximum observed improvements are around 0.6 on mR@20 (see the Scene Graph Detection column in Table 1 in [43]). CWS is complementary to the refinement process, and allows for more consistent graph generation within a step.
>
> $\textbf{3. Performance gains are not due to additional parameters.}$
>
> We argue that the improvements obtained by our refinement procedure are not a direct consequence of having an increased number of parameters. To demonstrate this further, we ran an additional experiment with a no-refinement transformer model that has a **similar number of parameters** as our proposed t=6 model in supplementary Table A1. The results are shown below. It is evident that our proposed refinement procedure allows for better learning, and the performance gains observed are not a result of having more parameters. We will add this ablation to the supplementary upon revision.
>
> | Model | mR@20/50 | R@20/50 | hR@20/50| AP | AP$_{50}$ | AP$_{75}$ | Model Size |
> |:----------:|:-------------|:----------:|:-------------:|:-------------:|:-------------:|:-------------:|:-------------:|
> | No Refinement  | 9.9 / 13.1 | 17.5 / 21.8 | 12.6 / 16.4 | 10.1 | 21.6 | 8.0 | 1.1x |
> | Refinement (t=6) | **11.8 / 15.8** | **21.0 / 26.1** | **15.1 / 19.7** | **14.6** | **27.6** | **13.2** | 1x |
>
> $\textbf{4. Originality and use of a decoder.}$
>
> Although we use a decoder as a part of our proposed transformer architecture, the primary contribution of our work is the iterative refinement formulation for scene graph generation, which can be realized using different architectures. We show this by additionally augmenting our proposed formulation to MOTIF [53], which is a completely different architecture compared to the transformer model in [2]. Additionally, the conditioning we use across our decoders is much richer than prior work, leading to better scene graphs.
>
> $\textbf{5. Finer details in Table 1.}$
>
> We appreciate the argument. Omitting finer details from Table 1, one can see that our improvements are **very significant** (#16 and #22 in Table 1; 3.7 higher on mR@50, 10.2 higher on R@50). The intent of the fine grained analysis is not to show model effectiveness, but rather to illustrate the behavior of our approach in different settings. Ideally, we would’ve put this in a separate table, but decided against it due to space constraints. Additionally, we apologize for the formatting in Table 1, and will make sure to highlight top performing entries for better readability.

---

> > ### Comment · Reviewer_h8U6 · 2022-08-08
> > **Response to Rebuttal**
> >
> > Thanks for the detailed response to my concerns. I am overall very satisfied with the response and will consider revising my score.
> >
> > - My apologies for missing Figure A1 completely during the review - this figure is what I was looking for and definitely boosted my confidence in the paper by a lot.
> >
> > - Thanks for the clarification on CAS & CWS. The explanation makes sense to me. Would be helpful to clarify the importance of the joint matching loss a bit more for the revision.
> >
> > - For the "no refinement" model: do they have similar number of layers as the Refinement(t=6) ones?
> >
> > - Thanks for clarifying the selection process of A3-A7. That is very good to know. Still, adding a bit more examples in the revision would be helpful (perhaps as a bunch of jpegs in the supplementary).

---

> > > ### Author Response · Authors · 2022-08-09
> > > **Response to Further Questions**
> > >
> > > We thank the reviewer for going through our rebuttal and are glad that we were able to address their concerns and questions. As suggested, we will clarify the importance of the joint loss in the revision and also add more examples in the supplementary.
> > >
> > > Regarding the "no refinement" model we present in the rebuttal, it has a similar number of parameters but only a **single** layer ($t=1$), that is we increase the number of parameters within each layer. Experiments pertaining to models having the same number of layers and parameters are already presented in the ablation study in **Table 2 (#2 and #3)**. #2 corresponds to the model with 6 decoder layers trained without any refinement, and #3 is the same 6 decoder layer model (equal number of parameters) but with our proposed joint loss (which enables refinement as explained previously). These two experiments highlight that increasing model capacity either via making each layer larger or via adding more layers (increasing model depth) does not emulate refinement. Therefore, our proposed refinement framework is necessary to obtain better performance.

---

### Official Review · Reviewer_Vh2W · 2022-07-13

**Rating:** 3
**Confidence:** 5
**Soundness:** 2 fair
**Presentation:** 2 fair
**Contribution:** 2 fair

**Summary:**

This paper propose an end-to-end scene graph generation framework, in which design iterative refinement manner to gradually optimize the predictions. Besides, it also introduce the reweighting loss to tackle the long tail problem in this task. Author verified this method in two popular dataset and achieve the superior performance. But this manuscript is not well-written and some expressions need to be improved.

**Questions:**

1. I have a question about the assumption in line 157.  The [1] had pointed out that the direction of the edge is important property in relationship prediction, but you assume a fixed information flow from subject to object and finally reach predicate. Please give the explaination of eliminating graph property in your assumption.
2. The proposed iterative refinement manner for SGG should involves the variation for both object and relationship, but figure 3 only present the estimation of different refinement steps for relationship.
3. The scaling parameters for reweighting loss is hard to set and your experiment in Table 1 also attempt various combinations, which present a relatively large impact. So, is there a more effective way to set these two parameters?
4. Author adopt the MOTIF [2] as baseline. As far as I know, this method use the prior information of dataset to bias the final predicted distribution. I would like to know that in your end-to-end framework, this kind of bias is still necessary for the your prediction?
[1]GPS-Net: Graph Property Sensing Network for Scene Graph Generation. In CVPR 2020.
[2]Neural Motifs: Scene Graph Parsing with Global Context. In CVPR 2018.

**Limitations:**

Scene graph is first proposed to use for image retrieval and it is also the more direct application compared with autonomous driving. So author should focus on some simple but effective applications for scene graph.

**Strengths And Weaknesses:**

#Strengths:
This paper reformulate the task of scene graph generation into an iterative optimization process. Based on transformer architecture, author expend the message passing within a Markov Random Field and design iterative refinement procedure. The performance in VG and AG dataset have achieve the state-of-the-art level.
#Weaknesses:
Although an end-to-end framework for generating scene graph is a good research direction, scene graph also involves object prediction. I think the pretrained object detector in two-stage methods will provide more better prediction, which authors did not given relevant introduction. The qualitative refinement analysis is not enough to support the effectiveness of the proposed method.

---

> ### Author Response · Authors · 2022-07-29
> **Response to Weaknesses and Questions**
>
> We appreciate the feedback from the reviewer, but would like to point out that most of the critiques stem from misunderstandings. We clarify these issues below and hope that the reviewer will carefully consider them.
>
> $\textbf{Two-stage detection will not provide better prediction. }$
>
> We respectfully disagree. As we discuss in lines 40 - 48 of the main paper, two-stage methods, although widely used in the scene graph generation literature [39, 45, 48, 50, 53], learn detectors that are oblivious to the graph generation task. Such object detectors are also slower (in terms of inference time) owing to their two-stage nature.   Additionally, as the object detector in these approaches is often pretrained, the identified objects have to be paired up before predicate assignment. Theseis leads to the need for the algorithm to consider quadratic number of such pairs, causing further inefficiencies in training and inference.
>
> To alleviate these issues, recent works in scene graph generation have adopted one stage architecture like transformers [13, 30] or convolutional networks [34]. Such one stage methods can be trained end-to-end and lead to faster inference. Although our proposed formulation in Equation 2 makes no assumption on the model architecture, we adopt a transformer based model owing to the aforementioned limitations of two stage architectures, and the overall trend in the literature.
>
> We compare against two stage methods in Table 1, and highlight that our proposed transformer based design outperforms existing approaches, including two-stage ones, by a large margin. Furthermore, we show the generality of our core approach by augmenting it to a two stage architecture in MOTIF [53] in Table 3, demonstrating that any existing method can leverage our proposed formulation for better scene graph generation.
>
> $\textbf{Questions}$
>
> $\textbf{1. Direction of edges is important for relationship prediction.}$
>
> We do believe that the direction of edges is critically important in predicate prediction, and our approach $\textbf{does not}$ eliminate this property. This is evident from Figure 3, where all generated graphs produced by our method have directed edges. Line 157 talks about the flow of information within a particular refinement step based on the factorization of the joint distribution (Equation 2). This flow of information arises from the use of chain rule, and does not imply that edge directions are ignored. The direction of the edge is implicitly encoded in the corresponding subject and object elements. In other words the direction of the edge is implicitly assumed to be going from $\texttt{subject}_i$ to $\texttt{object}_i$.
>
> As a simple example, consider an image consisting of two object instances -– a car and a pedestrian. The subject decoder would ideally identify both these objects in the image as potential “$\texttt{subjects}$”, e.g., “car” as $\texttt{subject}_1$ and “pedestrian” as $\texttt{subject}_2$. Conditioned on the “car” ($\texttt{subject}_1$) prediction, the object decoder will identify the “pedestrian” as its corresponding object ($\texttt{object}_1$), implying a directed edge that goes from car $\rightarrow$ pedestrian. The predicate decoder will then assign a class label to this edge, e.g., “\texttt{next to}” as $\texttt{predicate}_1$. Similarly, conditioned on the “pedestrian” prediction ($\texttt{subject}_2$), we get the directed edge pedestrian $\rightarrow$ car ($\texttt{subject}_2$).
>
> Important aspect of our model is that the three decoders are not independent and decoded instances among the three decoders are in correspondence with one another – forming a $\texttt{<subject, object, predicate>}$ triplet. The total number of such triplets is $n$ and corresponds to the number of decoded queries (which is the same for all three decoders).
>
> $\textbf{2. Proposed framework improves performance on both objects and predicates.}$
>
> Figure 3 (and more visualizations shown in supplementary figures A3-A7) highlight the iterative improvements of $\textbf{both}$ the object detection and predicate prediction tasks. The task of object detection involves identifying objects and simultaneously localizing them in an image. Figure 3 highlights better bounding box localization for the object corresponding to the class umbrella with each refinement iteration, which visually demonstrates improvement on the detection task. The figure is small, so this may be hard to see, but we would appreciate it if the reviewer can zoom in and examine the predicted object bounding boxes in addition to the graph itself.

---

> > ### Author Response · Authors · 2022-07-29
> > **Continued Response to Questions**
> >
> > $\textbf{3. Choosing the appropriate scaling parameter will depend on the underlying application. }$
> >
> > Choosing the appropriate scaling parameters is heavily influenced by the underlying application of the scene graph model. For example, in situations where long tail identification is important, setting higher alpha and beta values is desirable. We do not believe there is inherently a single choice that would be appropriate at all times and all applications. One of our contributions is highlighting this tradeoff by showing the performance of our model under different parameter values in Tables 1, 4 and supplementary table A2. Existing approaches lack the ability to modulate performance in such a manner, making their usability limited. Our approach, on the other hand, is able to handle a wider range of applications and allows users to set a desired tradeoff between effectively Recall and Mean Recall. We believe this to be a benefit and a desirable property as opposed to a limitation of our method.
> >
> > $\textbf{4. Proposed transformer model does not use any prior information. }$
> >
> > Our work presents a general framework for iterative refinement (Equation 2). We implement this framework using two different architectures -  a transformer based end-to-end model, and an existing method in MOTIF [53]. Our transformer based end-to-end network does not require the use of prior information (or bias) to generate scene graphs. All our results using this model are therefore devoid of this assumption.
> >
> > $\textbf{Limitations. }$
> >
> > Autonomous driving was mentioned only to highlight an important use case of scene graphs. Although there are plenty of other applications, our work focuses on the underlying algorithm of generating scene graphs, and is independent of the downstream task. Better scene graph prediction, which is the focus of this work, will improve all downstream applications of the predicted scene graphs.

---

### Meta-Review · Area_Chair_3KvR · 2022-08-29

**Recommendation:** Accept
**Confidence:** Certain

**Metareview:**

The authors propose a new approach for end-to-end training of predicting scene graphs from images (different from the traditional two-stage approach.) The key observation that the fixed factorization approach can be suboptimal due to error compounding is reasonable and is supported by the experiment results. The proposed solution with iterative refinement is reasonable and the design choices of the method are sound. Evaluation is comprehensive overall and the result is convincing.

Most of the key concerns raised by reviewers Vh2W and 5s37 who gave low scores (3 & 4) seem to be well addressed by the authors but the reviewers were not responsive and engaged in the follow-up discussion, so also no score update as well. However, the other reviewer h8U6 expressed that the authors addressed the concerns well and after reading the paper I agree on this based on my personal assessment. Therefore, even if two reviewers gave a rather low score 3 and 4, I cannot weigh those review scores much and rather rely more on the other reviewers giving 6 and my own assessment. So, I recommend accepting this paper even if the average score is rather lower than usual.

**Award:**

No

---

### Decision · Program_Chairs · 2022-09-14

Accept